# VISCON: Identifying and Benchmarking Vision Hallucination for Large Vision-Language Model

## Abstract

Large Vision-Language Models (LVLMs) have demonstrated exceptional capabilities in a variety of vision-language tasks, but suffer from "vision hallucinations" - a tendency generating text inconsistent with the image. This issue hampers their practical use in real-world applications. To effectively evaluate and detect these hallucinations, we introduce **VISCON** (VISual Concept cONsistency), a benchmark framework comprising a *benchmark image dataset* and *quantitative evaluation pipelines* to assess vision hallucinations in LVLMs. VISCON extends beyond previous hallucination metrics by offering: a) diverse image styles across multiple visual domains, b) evaluation of a broader range of visual concepts, including objects, attributes, and relationships, and c) high annotation density from detailed scene-graph annotations to reduce false negatives. These improvements enable comprehensive analysis of hallucinations related to both domain shifts and concept types and offer more accurate hallucination evaluation. To detect vision hallucinations, we propose two innovative evaluation pipelines within VISCON: an Earth Mover's Distance (EMD)-based pipeline and an "Evaluate-By-Edit" pipeline. The EMD-based pipeline measures the distributional similarity between the reference visual concepts and those mentioned by LVLMs, robust against vocabulary shifts between annotations and natural language responses. The "Evaluate-By-Edit" focuses on the edit distance between the original LVLM response and a hallucination-reduced version revised according to the rich visual concept annotations, providing an interpretable analysis of hallucinated content. Importantly, our method directly evaluates captioning responses, unlike previous metrics that query the existence of individual visual concepts. This approach is more challenging, as it requires models to handle multiple concepts simultaneously, providing better discrimination of LVLM performance. Through extensive experiments on six leading LVLMs, VISCON reveals crucial insights into the nature of vision hallucinations. Our findings indicate that factors such as image domain shifts, complexity of visual concepts and model response length significantly influence the occurrence of hallucinations in LVLM responses. Additionally, human evaluations confirm that VISCON aligns with human preferences better than established hallucination metrics.

## 1 Introduction

Large Vision-Language Models (LVLMs), which integrate Large Language Models (LLMs) with visual perception capabilities, have made significant strides towards developing generalist AI systems. Models like GPT-4V(OpenAI, 2023), LLaVA(Liu et al., 2023c), BLIP-2(Li et al., 2023a), InstructBLIP(Dai et al., 2023), and Fuyu(Bavishi et al., 2023) have showcased exceptional performance in a range of vision-language tasks. However, a pivotal challenge that these LVLMs face is the phenomenon of "vision hallucinations" (Li et al., 2023b; Liu et al., 2023a), where they generate responses inconsistent with the image. As shown in Figure 1, even the leading LVLMs like GPT-4V and LLaVA-1.5 produce errors misrecognizing visual concepts (colored text), serving as a proof of the importance to evaluate and investigate the vision hallucinations.

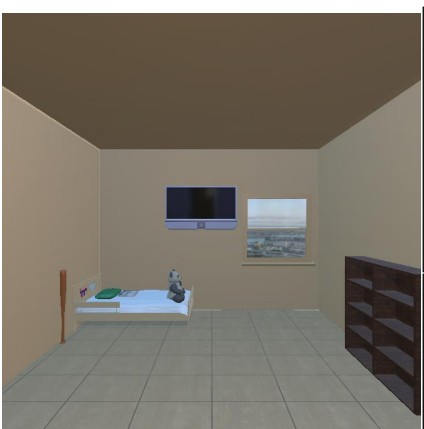

**GPT-4V:**
This image appears to depict a simple 3D rendered room. The walls are a light beige color with a slightly darker ceiling, and the floor has a plain tile texture. On the left, there is a basic bed with a wooden frame and a blue cover featuring star patterns; a gray plush toy is placed on the bed. Mounted on the wall above the bed is a flat screen television. On the right side of the room, there is a window providing a view of a landscape with buildings, which appears to be more realistically textured compared to the simplicity of the room's design. Next to the window is a dark brown chest of drawers without any items placed on top of it. Overall, the room has an uncluttered, minimalist aesthetic. There are no people present in the room. The minimalism and the contrast in rendering style between the room and the view outside give the image a somewhat surreal quality.

**LLaVA-1.5:**
The image depicts a small, clean room with a bed positioned in the corner. The room is painted white and features a television mounted on the wall above the bed. A teddy bear is placed on the bed, adding a touch of warmth to the space. In addition to the bed and television, there is a book on the bed, possibly for reading before sleep. A chair is also present in the room, located near the left side of the bed. The room appears to be a cozy and comfortable space for relaxation and rest.

Figure 1: Vision hallucinations of LVLMs. Models tend to generate responses inconsistent with the image. We show results from GPT-4V and LLaVA-1.5, the state-of-the-art private/open-source LVLMs. Hallucination text about objects (category), attributes, and relationships is marked as red, green and orange repectively.

To effectively address vision hallucinations and improve LVLMs practicality in real-world applications, robust metrics are essential. Existing methods like POPE (Li et al., 2023b) and CIEM (Hu et al., 2023) assess hallucinations by querying models about visual concepts, providing useful insights but with notable limitations. POPE focuses solely on object categories, overlooking attributes and relationships, which our empirical evidence (Figure 1, Table 4) suggests are also prone to hallucination. CIEM relies on image captions, which suffer from human reporting bias. Both methods suffer from incomplete visual concept coverage, leading to potential false negatives. Additionally, querying individual concepts one by one is too simplistic for large-scale pretrained LVLMs, limiting their ability to fully assess the model's performance on more complex, multi-concept tasks. HallusionBench (Liu et al., 2023a) offers qualitative analysis but lacks a quantitative metric. Additionally, none of these methods evaluate hallucinations across a broad range of image domains.

To address these deficiencies, we present **VISCON** (VISual Concept cONsistency), a comprehensive vision hallucination benchmark framework comprising a *benchmark image dataset* and *quantitative evaluation pipelines*. VISCON advances prior metrics by: a) incorporating diverse image domains, b) evaluating a broad range of visual concepts (objects, attributes, and relationships), thus extending to a comprehensive vision spectrum in evaluation and c) leveraging very dense scene-graph annotations to reduce false negatives and improve accuracy. Additionally, our method directly evaluates captioning responses rather than querying each individual visual concepts, requiring LVLMs to handle multiple visual concepts together, offering a more rigorous and discriminating assessment.

Our *benchmark image dataset* is meticulously curated, incorporating real-world images from the Visual Genome dataset (Krishna et al., 2017), known for its rich scene-graph annotations of diverse objects, attributes, and relationships. VISCON also includes 3D-rendered images from the PROC-THOR dataset (Deitke et al., 2022), selected for its extensive and easily accessible scene-graph annotations, offering a more scalable and complete perspective on visual concepts. Furthermore, VISCON broadens the scope of evaluation to include various image domains, such as different styles (cartoon, sketch, etc.) and sources (real-world and 3D-rendered), facilitating the analysis of impact of vision domain shifts on hallucination. Upon acquiring a diverse set of probe image, we construct reference visual concept set for each image from their scene-graph annotations, serving as a gold standard to compare LVLM's output with. While we recognize that annotations, particularly for attributes and relationships, may not be entirely exhaustive, our dataset provides significantly more comprehensive coverage compared to previous hallucination benchmarks or captioning metrics (Table 1).

For *quantitative hallucination evaluation*, we propose two pipelines: an Earth Mover's Distance (EMD)-based pipeline and a novel "Evaluate-By-Edit" pipeline. The EMD-based pipeline measures the distributional similarity between referential visual concepts present in image and the LVLM's mentioned concepts, providing **robust assessment against vocabulary shifts** between scene-graph annotations and natural LVLM responses. This objective approach is complemented

by the "Evaluate-By-Edit" pipeline, which enhances **interpretability on detailed hallucinated content**. It assesses hallucinations by calculating the edit distance between the LVLM's original response and a revised version with reduced hallucinations, refined through a query-and-revise process. This revision process uses an LLM to generate queries based on objects, attributes, and relationships mentioned in the original text, and refines the response using query results based on the reference visual concept set. The combination of these pipelines ensures a comprehensive, robust and interpretable evaluation of vision hallucinations in LVLMs.

We apply VISCON to six leading models including LLaVA-1.5(Liu et al., 2023b), InstructBLIP(Dai et al., 2023), Fuyu(Bavishi et al., 2023), Qwen-VL (Bai et al., 2023), Phi-3V (Abdin et al., 2024) and GPT-4V(OpenAI, 2023), conducting comprehensive evaluations. Our quantitative experiments evaluate vision hallucinations of model responses from LVLMs, and analyze the relationship between vision hallucination and multiple factors, including image domains and their domain shifts, visual concept type, and model response length. With empirical evidences, VISCON aligns more closely with human preferences than existing hallucination metrics. These findings provide significant insights into the nature of vision hallucinations in LVLMs and offer methodologies for their effective assessment.

In our research, we present the following key contributions: 1) we introduce **VISCON** (VISual Concept cONsistency), a unique metric for assessing vision hallucinations in large vision-and-language models (LVLMs). VISCON stands out by its wide range annotations of objects, attributes, and relationships from easily-accessed image scene-graphs, comprehensive scope of tested images with diverse sources and styles, vocabulary-shift robust evaluation with proposed EMD-based pipeline and interpretable hallucination evaluation with "Evaluate-By-Edit" pipeline. 2) we conduct extensive quantitative and qualitative experiments to explore the phenomenon of vision hallucinations in a wide range of popular LVLMs, including LLaVA-1.5, InstructBLIP, Fuyu, Qwen-VL, Phi-3V and GPT-4V. Through these experiments conducted using VISCON, we not only highlight the prevalence of hallucinations in current models but also uncover and analyze various factors that contribute to hallucinations in LVLM responses. 3) we validate VISCON through human evaluation, demonstrating its enhanced alignment with human judgments compared to existing metrics, which is key to evaluate free-form LVLM responses.

## 2 RELATED WORK

**Large Vision-Language Models (LVLMs) and Vision Hallucination:** In the quest for versatile artificial intelligence, Large Vision-Language Models (LVLMs) aim to equip powerful Large Language Models (LLMs) with visual comprehension. These models have demonstrated state-of-the-art performances in zero-shot and fine-tuned scenarios, such as GPT-4(OpenAI, 2023), BLIP-2(Li et al., 2023a), InstructBLIP(Dai et al., 2023), LLaVA(Liu et al., 2023c) and Fuyu(Bavishi et al., 2023), etc. Despite advancements, vision hallucination remains a critical issue, with models often producing image descriptions that misalign with the actual contents. This undermines LVLM reliability, especially in tasks like image captioning, questioning both their accuracy and real-world applicability. Our metric contributes to addressing this challenge, by detecting and evaluating such hallucinations. We construct a comprehensive visual concept set derived from image, including objects, attributes and relations present in image, serving as a gold standard against which model outputs can be evaluated.

**Vision Hallucination Metrics:** Assessing vision hallucinations in LVLMs is pivotal for their practical deployment. While metrics for linguistic hallucinations in LLMs are well-established (Chen et al., 2023)(Yang et al., 2023)(Dong et al., 2023), evaluating vision hallucinations presents unique challenges. Existing LVLM metrics, such as POPE (Li et al., 2023b) and CIEM (Hu et al., 2023), are limited in scope. POPE focuses only on object presence, ignoring attributes and relationships, while CIEM relies on incomplete visual concepts from captions due to human reporting biases. AMBER(Wang et al., 2024) extends hallucination evaluations towards attributes and relations, but only annotate scarse visual concepts from image, leading to potential false negatives. These methods adopts a VQA format, which simplifies the task by querying individual concepts rather than assessing the model's ability to handle multiple concepts in free-form responses. HallusionBench (Liu et al., 2023a) provides qualitative insights but lacks quantitative metrics. VISCON overcomes these limitations by leveraging dense scene-graph annotations for a comprehensive evaluation of halluci-

nations. It assesses free-form descriptions rather than individual existence query of visual concepts, reflecting real-world LVLM use, and covers diverse image domains. Human studies confirm VISCON's alignment with human preferences, underscoring its efficacy and applicability.

# 3 VISCON (VISUAL CONCEPT CONSISTENCY)

## 3.1 BENCHMARK IMAGE DATASET

VISCON evaluates LVLMs' hallucinations based on two sets of base images: (1) real-world images from the VisualGenome dataset (Krishna et al., 2017) and (2) 3D-rendered views of indoor rooms from the PROCTHOR dataset (Deitke et al., 2022). The scene-graph data for real-world images were sourced from VisualGenome annotations, while for 3D-rendered images, the scene-graphs were generated based on PROCTHOR's object position and attribute annotations. We adopt 3D-rendered images because the 3D scenes in PROCTHOR provides a relatively complete annotaions of visual concepts, making these images and related scene-graph easy to acquire and easy to scale to a large quantity.

**Evaluate Hallucination against Domain Shift** To investigate the relationship between vision hallucinations and image domain shifts, we supplement our analysis by extending image splits beyond the real-world images from VisualGenome(Krishna et al., 2017) dataset, as shown in Figure 2a. We focus on two distinct types of domain shifts to test vision hallucinations: 1) *Real-World vs. 3D-Rendered*: We examine hallucinations in 2D views of simulated 3D indoor scenes from PROC-THOR(Deitke et al., 2022), noting domain shifts like unusual textures. 2) *Real-World vs. Stylization*: Both real-world and 3D-rendered images are transformed into various visual styles — sketch, line painting, cartoon, and oil paint — to assess the impact of style-based domain shifts. Through benchmark on these image domains, VISCON provides crucial insights into their performance across diverse visual domains.

**Scene-Graph Based Visual Concept Extraction and Reference Set Construction** Upon acquiring a diverse set of probing image set for hallucination evaluation across different image sources and domains, we meticulously assemble a comprehensive visual concept reference set. This set includes objects, attributes, and relationships extracted from scene-graphs, providing broader coverage than traditional object annotations or caption-based methods. Our goals are twofold: a) to provide denser and more complete visual concept annotations, minimizing false negatives in hallucination evaluations, and b) to extend hallucination evaluation to a wider range of visual concepts, enabling more comprehensive assessments.

For real-world images, we utilize a subset of annotated scene-graphs from the VisualGenome dataset (Krishna et al., 2017). For 3D-rendered images, we source views of indoor rooms from the PROCTHOR dataset (Deitke et al., 2022), generating scene-graphs automatically from these 3D scenes with pre-defined rules. This process ensures a diverse and extensive reference set, spanning various image domains and capturing a wide array of visual concepts, including objects, attributes, and relationships.

Table 1: Comparison of average annotation density of visual concepts. *: estimated from subset.

| Split | Objects | Attributes | Relations |
|---|---|---|---|
| Ours (real-world) | 23.0 | 30.3 | 19.4 |
| Ours (3D) | 17.0 | 8.3 | 53.7 |
| POPE | 3.2 | - | - |
| AMBER | 4.9 | 7.6 | 1.7 |
| MSCOCO* | 3.2 | 4.9 | 3.9 |

More specifically, for each image, we extract a set of object names, attribute pairs (e.g., object A and attribute X) and relationship triplets (e.g., object A, subject B and relationship Y) from its scene-graph annotations.

In result, our reference set achieves significantly higher annotation density compared to previous hallucination and captioning metrics. As shown in Table 1, our visual concept annotation density surpasses that of POPE (an established hallucination metric) and MSCOCO (a representative captioning metric). By leveraging scene-graph annotations from VisualGenome and object metadata from 3D simulators, VISCON extends hallucination evaluation to a broader range of visual concepts and reduces false negatives compared to existing metrics.

## 3.2 HALLUCINATION METRICS

We propose two innovative metrics, Evaluate-By-EMD and Evaluate-By-Edit, to assess vision hallucinations in LVLMs with aforementioned diverse probe image set and constructed reference set of visual concepts. In principle, EMD and Evaluate-By-Edit are complementary metrics with distinct focus and functionality: a) EMD-based pipeline is designed to provide a **quantitative assessment** that is robust to vocabulary shifts. In operation aspect, it compares two visual concept sets: one from annotations and another from model responses. By embedding all visual concepts into a language embedding space, EMD calculates the minimum transport cost from the model's mentioned set to the annotated reference set. This approach is robust to vocabulary shifts, as synonyms typically have similar embeddings. b) Evaluate-By-Edit pipeline is designed to offer **hallucination evaluation interpretability** by enabling visualization of text edits made during the hallucination cleaning process. In operation aspect, It calculates the edit distance between two captions: the original model response and a refined, hallucination-reduced version. The revision is done by querying the existence of mentioned visual concepts and revising them according to query results. This allows for visualization and interpretation of hallucinations by showing the edited parts.

### 3.2.1 EVALUATE-BY-EMD

As depicted by Figure 2, our "Evaluate-by-EMD" pipeline evaluates LVLMs by comparing the visual concepts they mention with a pre-established reference set from image scene-graphs. This pipeline consists of two stages:

**Extraction of Visual Concepts from LVLM Output**: As illustrated in Figure 2b, we first prompt the LVLM to describe the image in detail. Using GPT-4 with few-shot prompts, we extract visual concepts (objects, attributes, relationships) from the model's response. These concepts form a set representing the model's interpretation of the image, in the same format as the constructed reference set (i.e., a set of object names, attribute pairs and relationship triplets). Some LVLMs, like GPT-4V, often produce vague responses (e.g., "a remote control or smartphone") to avoid inaccuracies. Our visual concept extraction process, however, extracts both options from such expressions, penalizing this "smart" strategy and providing a more accurate measure of the model's real hallucination level.

**Vocabulary Shift-Robust Hallucination Evaluation with EMD:** In comparing two set of visual concepts, either from reference or model output, we confront the challenge of vocabulary shift — the variance between the scene-graph annotation and the LVLM's open-ended text generation (e.g., "dog" vs. "puppy"). To reconcile this ambiguity, we compute the Earth Mover's Distance (EMD) between textual embeddings of visual concepts from the LVLM and the reference set instead of exactly matching them word by word. We use a pretrained sentence encoder $E$ to generate embeddings for each visual concept $t \in \{\text{object}, \text{attribute}, \text{relation}\}$ and for both the reference set of $\mathcal{V}^t = \{v_1^t, ..., v_N^t\}$ and the model-mentioned set $\mathcal{W}^t = \{w_1^t, ..., w_M^t\}$: $F_{\mathcal{V}^t} = \{E(T(v_i^t)), ..., E(T(v_N^t))\}, F_{\mathcal{W}^t} = \{E(T(w_i^t)), ..., E(T(w_M^t))\}$. Here $T$ is the mapping that fits each visual concept into corresponding template. For instance, the attribute pair (couch, brown) is encapsulated as "attribute of couch: brown", and the relation triplet (couch, besides, wall) is formatted as "relation: couch besides wall". The hallucination metric is the sum of EMDs across all visual concept types:

$$d_{\text{EMD}} = \sum_t d_{\text{EMD}}^t, t \in \{\text{object}, \text{attribute}, \text{relation}\},$$

$$d_{\text{EMD}}^t(F_{\mathcal{V}^t}, F_{\mathcal{W}^t}) = \min_{U \in \Gamma} \langle U, C_{\mathcal{V}^t, \mathcal{W}^t} \rangle_F \tag{1}$$

Here $U \in \mathbf{R}^{N \times M}$ is an transporation matrix between two sets of embeddings, $\Gamma$ is the set of all possible transportation matrices, $C_{\mathcal{V}^t, \mathcal{W}^t} \in \mathbf{R}^{N \times M}$ is the element-wise cosine dissimilarity matrix. The optimal transport matrix can be seen as a set of soft matching relationships between two set of of visual concepts, and a smaller distance $d_{\text{EMD}}$ signifies fewer vision hallucinations.

### 3.2.2 EVALUATE-BY-EDIT

As depicted in Figure 3, our "Evaluate-by-Edit" pipeline revises LVLM responses using a visual database constructed from image scene-graph annotations, then evaluates hallucination by calculating the edit distance between the original and revised text. This pipeline consists of three stages:

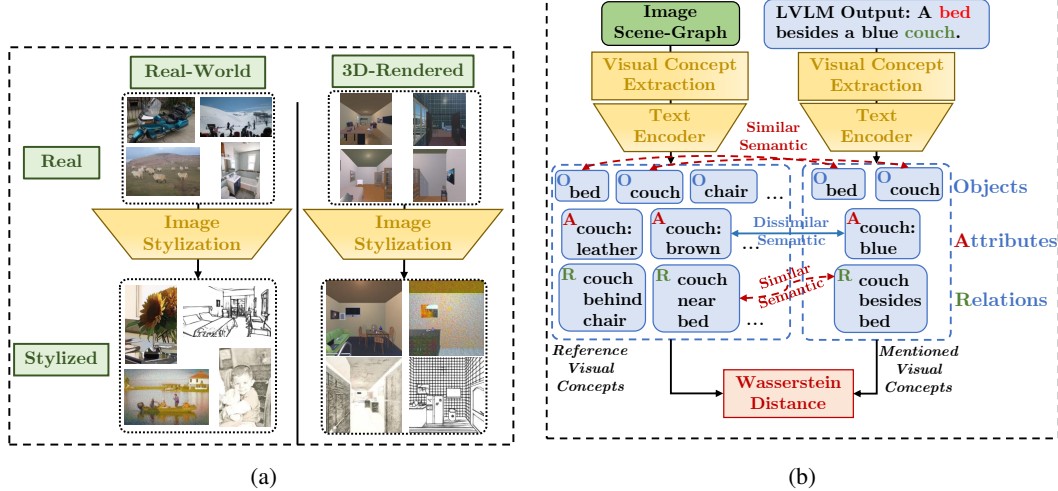

(a) (b)

Figure 2: **(a)**. The VISCON metric evaluates LVLMs across various visual domains to analyze robustness against visual domain shifts. We acquire these additional images from either 3D rendering or image stylization. **(b)**. Overview of the Earth Mover's Distance (EMD)-based pipeline in VISCON. Visual concepts are extracted from image scene-graphs to create a reference set and from model responses to form a mentioned set. Textual embeddings of these concepts are used to compute EMD, effectively handling vocabulary shifts.

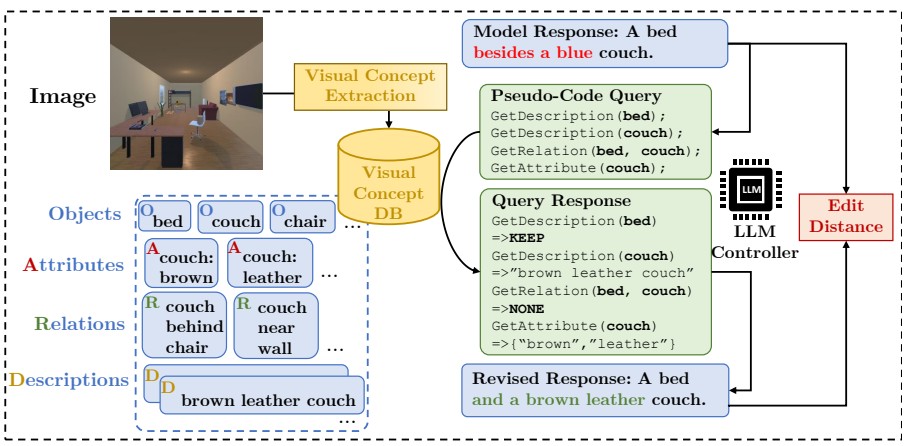

Figure 3: Overview of the "Evaluate-By-Edit" pipeline in VISCON. A rich visual concept database from scene-graphs is queried using pseudo-code generated by an LLM to revise the model response. The edit distance between the original and revised responses measures hallucinations.

**Constructing Database for Response Refinement:** Initially, we construct a comprehensive visual concept database to serve as a foundation for model response revision and hallucination validation, as shown in left part of Figure 3. We first include reference visual concept set as constructed in Section 3.1, and additionally enrich with manually annotated object and region-level descriptions to provide a detailed context for later revision procedure. For of 3D-rendered images, we further include spatial locations for each visible object, allowing for precise validation of spatial relations.

**Visual Query Generation and Execution:** Using GPT-4, we generate pseudo-code queries to validate visual concepts mentioned by the LVLM, as shown in right part of Figure 3. These queries verify the presence, attributes, and relationships of objects. For example, a query might confirm whether a described object actually exists in the image. In the query generation, we tailor the queried object names align with the database's vocabulary, to ensure relevance and precision in verification. By executing database search, each visual concept related query results in "NONE" (non-existent), "KEEP" (confirmed or inconclusive), or specific textual descriptions (confirmed existence and have database-sourced captions). In principle, to reduce hallucination, we only remove visual concepts in

Table 2: Comparison on the Earth Mover's Distance (EMD) metric of different LVLMs. "Human Annotation" refers to the performance of annotated caption. Best and second best LVLM performances are marked **bold** and underlined.

| Model | mean | EMD↓ | | | | | | | | | |
| --- | --- | --- | --- | --- | --- | --- | --- | --- | --- | --- | --- |
| | | real-world | | | | | 3D | | | | |
| | | original | cartoon | sketch | oil painting | line | original | cartoon | sketch | oil painting | line |
| InstructBLIP | 106.8 | 105.0 | 105.8 | 106.1 | 105.5 | 108.7 | 106.3 | 107.5 | 108.1 | 107.2 | 107.9 |
| LLaVA-1.5 | 105.5 | 104.4 | 105.1 | 105.8 | 105.9 | 108.9 | 103.6 | 104.8 | 105.0 | 105.4 | 106.2 |
| Fuyu | 107.7 | 104.8 | 104.7 | 110.8 | 108.4 | 111.8 | 105.1 | 105.3 | 108.0 | 107.1 | 110.8 |
| Qwen-VL | 106.3 | 103.5 | 104.9 | 106.8 | 106.5 | 111.8 | 102.5 | 103.3 | 106.3 | 106.0 | 111.1 |
| Phi-3V | 105.2 | **100.3** | 102.6 | 105.4 | 105.0 | 110.7 | 103.5 | 103.2 | 106.6 | 105.3 | 109.2 |
| GPT-4V | **103.0** | 101.3 | **102.0** | **104.2** | **104.5** | **108.1** | **100.1** | **100.6** | **102.6** | **100.6** | **105.7** |
| Human Annotation | | 101.3 | | | | | 98.8 | | | | |

Table 3: Edit distances between source and revised captions for different LVLMs. We perform linear regression between source caption length and total edit distance, with slopes denoted as $\alpha$. Best and second best performances are marked **bold** and underlined.

| Model | Edit Distance | | $\alpha$ |
| --- | --- | --- | --- |
| | total | per-word | |
| Fuyu | 72.5 | 0.85 | 1.10 |
| InstructBLIP | 83.1 | 0.83 | 0.98 |
| LLaVA-1.5 | 59.2 | 0.69 | 1.14 |
| Qwen-VL | 56.0 | 0.76 | 0.95 |
| Phi-3V | 47.2 | 0.62 | 0.79 |
| GPT-4V | **44.1** | **0.59** | 1.08 |

captions that are non-existent in visual concept database, and modify text with conflicts with short object descriptions. For detailed query types and prompts, please refer to appendix.

**Response Refinement and Evaluation with Edit Distance:** Based on query results, we refine the LVLM's original responses $T_{\text{original}}$ to produce a revised version $T_{\text{revised}}$. Non-existent concepts are removed, conflicts are modified according to query results, and visual concepts with valid or inconclusive query results are retained. We then calculate the word-based edit distance $d_{\text{EDIT}}$ between the $T_{\text{original}}$ and $T_{\text{revised}}$ as a measure of hallucinations using both total edit distance ($d_{\text{EDIT}}$) and per-word edit distance ($d_{\text{EDIT}}/|T_{\text{original}}|$) as metric. The edit distance metrics provides a direct and interpretable assessment of hallucinated content, allowing users to identify specific hallucinated parts. Despite our efforts to collect dense visual concept annotations, some queries may remain inconclusive and unchanged. However, these cases are rare, and even when they occur, the edit distance still provides a reliable lower bound for estimating hallucinations. Despite potential vocabulary shift issues, it complements EMD-based evaluation by offering an interpretable method to assess visual hallucinations. As a by-product, this method yields a cleaned version of the model's output without requiring additional LVLM training or being specific to any LVLM.

# 4 ANALYSIS

## 4.1 EXPERIMENT DETAILS

Six representative LVLMs, namely LLaVA-1.5, InstructBLIP, Fuyu, Qwen-VL, Phi-3V, and GPT-4V (`gpt-4-1106-vision-preview`), were evaluated across 10 image domains from our probe image set. Responses were generated using default nucleus sampling. For EMD, the metric was scaled by 100 for comparison. Edit distance metrics were calculated by comparing the total and per-word edit distances between the original and revised model responses.

## 4.2 ANALYSIS OF HALLUCINATION

**Hallucinations for Different LVLMs** Our comparative analysis of EMD distances (Table 2) and edit distances (Table 3) reveals that GPT-4V experiences the least visual hallucination, as indicated by its minimal mean-EMD and two edit-distance related axis intercepts. Newer model like Phi-3V

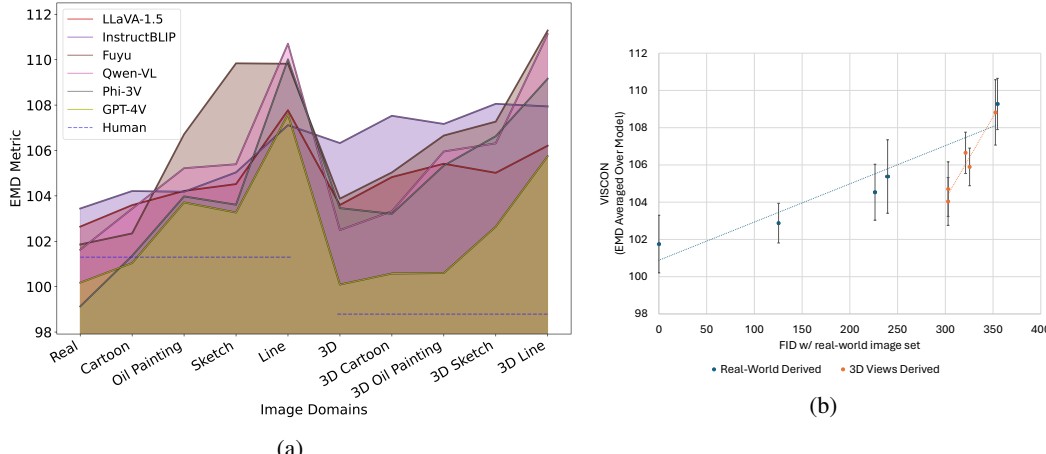

(a)

(b)

Figure 4: **(a)**. Comparison of EMD (y-axis) vs. image domain (x-axis). We observe more severe hallucinations with increasing domain shifts from real-world images, and more hallucination for 3D rendered than real-world images. **(b)**. Comparison of EMD (y-axis) vs. image domain FID (x-axis, compared to real-world images). There is a clear positive correlation (Pearson r = 0.83) between image domain FID and the EMD of LVLM responses.

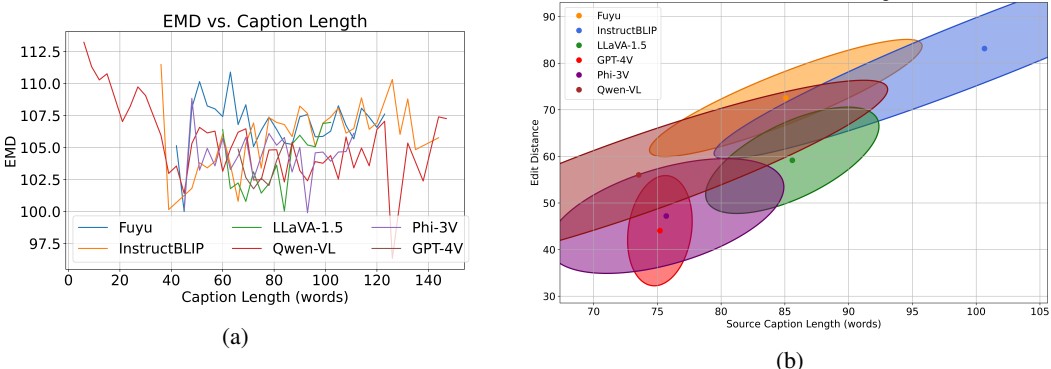

(a)

(b)

Figure 5: **(a)**. Comparison of EMD (y-axis) vs. response (caption) length (x-axis). **(b)**. Total edit distance (ED) between source and fixed caption (y-axis) vs. source caption length (x-axis, model output length) of different LVLMs. For clearer visualization, we plot the estimated Gaussian distribution of the joint distribution of edit distance and caption length, represented as ellipses where the radii correspond to the covariance directions. We observe similar EMD across different output length and linear increasing edit distance as the response length increases, which suggests longer model responses hardly provide more accurate information.

shows performance closely aligned with GPT-4V, especially with fewer hallucinations in real-world images. Interestingly, GPT-4V performs relatively bad on very abstract images (line painting splits) despite its impressive authenticity on other image splits, which maybe due to the tendency of GPT-4V to generate safe but vague responses (e.g., "it is A or B") that are penalized by VISCON.

When comparing human performance (using annotated captions), it is evident that all models underperform relative to human on images with domain shifts, while some model (GPT-4V and Phi-3V) can achieve similar or even better performances on original real-world images, which may due to the extensive pretraining of these LVLMs and rare but existing incorrectness in human annotations.

As shown in Table 3, GPT-4V demonstrates the lowest total and per-word edit distances, indicating its superior performance in minimizing hallucinations. Phi-3V closely follows, reflecting similar trends in EMD evaluations, suggesting that newer LVLM models are gradually approaching GPT-4V's impressive performance.

**Impact of Visual Domain Shift on Hallucination** The EMD metrics of the LVLMs across different visual domains (Table 2) and the trend of EMD vs. image domains (Figure 4a) reveal several

insights: a) all evaluated LVLMs perform relatively well on real-world images, likely due to their pretraining on similar image distributions. b) image domain shifts in the visual domain due to image stylization consistently introduce more hallucinations, both in real-world and 3D-rendered images.

To quantatively showcase the impact of image domain shifts, we calculated the Fréchet Inception Distance (FID) across different image domains and explored its correlation with EMD evaluations. Figure 4b shows the mean EMD compared to image domain FID, with error bars representing the EMD's sample-wise standard deviation across image domains. As the image domain diverges from real-world imagery, models exhibit higher levels of hallucination. Additionally, the wide range of domain gaps, from 125.3 (real-world, cartoonized) to 352.5 (3D view, line painting), and their relationship with the EMD metric, validate the effectiveness of our image probe set in showcasing LVLM hallucinations.

Table 4: EMD vs. visual concept type (all, object, attribute or relation). Best and second best performances are marked **bold** and underlined. We observe LVLMs handles objects more accurately.

| Model | EMD↓ | | | |
|---|---|---|---|---|
| | object | attribute | relation | all |
| InstructBLIP | 29.40 | 41.55 | 35.15 | 106.10 |
| LLaVA-1.5 | 28.79 | 41.43 | 34.56 | 104.78 |
| Fuyu | 29.71 | 41.66 | 35.65 | 107.03 |
| Qwen-VL | 28.98 | 41.41 | 35.17 | 105.56 |
| Phi-3V | 28.69 | 40.85 | 35.05 | 104.59 |
| GPT-4V | **27.63** | **40.42** | **34.50** | **102.54** |

Empirically, we observe that edit distance metrics are significantly influenced by the possible additional text describing the image styles (e.g., "this image appears to be a sketch of ..."). Such stylistic texts also impact edit distances, introducing confounders to the comparison across visual domains using edit distance metrics. Therefore, our analysis primarily focuses on examining the impact of domain gaps by comparing EMD evaluations that ignores these texts.

**Relationship Between Hallucination and Visual Concept Type** As we can distinguish three types of visual concepts in EMD calculation, we can compare the severity of hallucinations for different visual concept types (Table 4). It becomes evident that objects are the most accurately represented concepts, and attributes and relations are the more prone to hallucinatory output. This trend can be attributed to the increasing complexity of representing more elements from object to attributes and relations. All tested models consistently generated less accurate descriptions for attributes and relations than for objects across all visual domains (Table 4), underscoring the challenge in accurately processing more complicated visual concepts.

**Relationship Between Hallucination and Model Response Length** Consistent EMD

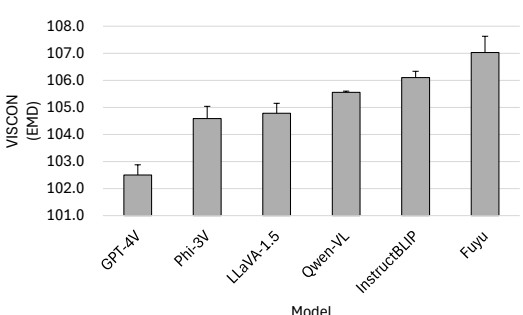

Figure 6: EMD v. vocabulary shifts. We show the EMD changes after synonym replacements as error bars. We observe minor resulting changes, validating the stability of EMD against vocabulary shifts.

values across varying text lengths (Figure 5a) suggest that longer outputs do not provide more accurate information. Visualizing total edit distance against response length (Figure 5b) reveals a clear linear relationship. Linear regression confirms that total edit distance increases almost 1-to-1 with output length (slope $\alpha$ in Table 3), indicating that additional content in longer outputs is mostly removed during revision.

## 4.3 ANALYSIS OF VISCON

**Correlation to Human Evaluation** We compared our VISCON metric with the established POPE metric through human evaluations of randomly selected responses subset from various models. Experts rated hallucinations on a 1 to 5 scale, with 5 indicating minimal hallucination. The POPE metric assesses hallucination tendencies by querying object existence using various selection strategies (random, popular, adversarial). In Table 5, our analysis showed GPT-4V favored by both VISCON and human assessments, while POPE favored LLaVA-1.5. This discrepancy is due to POPE's dif-

Table 5: Comparison between VISCON (EMD-based and QA-based) and previous LVLM hallucination metric, namely POPE, and their correlation strength to human preference (mean $\pm$ std). Largest and second largest correlation coefficients are marked **bold** and underlined. [†]: Evaluated with 10% question data, due to the high cost of GPT-4V.

| Method | POPE↑ | | | VISCON-QA (Acc↑) | VISCON-EMD↓ | AMBER↑ | VL-Task Performance VQAv2↑ | Human Eval Score↑ |
|---|---|---|---|---|---|---|---|---|
| | R | P | A | | | | | |
| GPT-4V | 71.0 | 73.8 | 73.4 | 76 [†] | 103.0 | 91.4 | 77.2 | 3.90 $\pm$ 1.11 |
| Phi-3V | 69.1 | 66.7 | 64.8 | 62 | 105.2 | - | - | 2.73 $\pm$ 1.18 |
| Qwen-VL | 70.7 | 72.4 | 69.8 | 50 | 106.3 | 89.7 | 78.2 | 2.73 $\pm$ 1.34 |
| LLaVA-1.5 | 77.6 | 74.3 | 79.2 | 72 | 105.5 | 83.5 | 78.5 | 2.47 $\pm$ 1.06 |
| InstructBLIP | 71.9 | 68.0 | 70.2 | 22 | 106.8 | 86.5 | - | 2.21 $\pm$ 1.02 |
| Fuyu | 61.0 | 57.3 | 61.9 | 46 | 107.7 | - | 74.2 | 2.03 $\pm$ 1.13 |
| Kendall-$\tau$ | 0.14 | 0.41 | 0.28 | 0.55 | **0.83** | 0.80 | 0.00 | - |
| Spearman | 0.12 | 0.49 | 0.35 | 0.75 | **0.90** | 0.67 | 0.20 | - |
| Pearson | 0.26 | 0.61 | 0.35 | 0.66 | **0.95** | 0.77 | 0.38 | - |

ficulty in interpreting GPT-4V's nuanced outputs which often extend beyond simple 'yes' or 'no' answers. VISCON excels by accurately identifying visual concepts in text. Moreover, we computed Kendall-$\tau$, Spearman, and Pearson correlation coefficients to compare metrics with human judgments. Higher coefficients indicate better empirical correlation. We also compared EMD-based evaluation (VISCON-EMD) with: model performances on vision-language tasks like VQAv2, and VISCON-QA (see Appendix D.4 for metric details), a QA-based hallucination evaluation method using VISCON's visual concept reference set but querying each visual concept individually. In Table 5, while QA-based metrics, such as VISCON-QA and POPE, are useful for assessing specific recognition capabilities, they fall short compared to EMD-based methods. EMD-based evaluations are more effective because they 1) assess the entire descriptive response, capturing a broader range of hallucinations, and 2) require models to handle multiple visual concepts simultaneously, increasing the likelihood of hallucinations and better revealing model capabilities. As a result, VISCON (EMD-based) showed the highest correlations with human evaluations, confirming its accuracy and discriminating power, and demonstrating that QA-based metrics or VL tasks alone does not fully capture hallucination severity.

**EMD Stablity Against Vocabulary Shift** We assess the EMD metric's stability to vocabulary changes by substituting synonyms in LVLM responses. We prompt an LLM to replace one or two words with synonyms to simulate these shifts. Figure 6 shows EMD distances without word replacements as columns, with perturbation ranges as error bars. Results indicate minimal EMD changes ranging from 0.04 to 0.60, and synonym replacements do not affect the comparative rankings among evaluated LVLMs, except two already very closely performing model (Phi-3V and LLaVA-1.5). This confirms EMD's robustness in evaluating LVLM outputs despite vocabulary variations.

# 5 CONCLUSION

In this study, we introduce **VISCON**, a novel vision hallucination benchmark framework designed for LVLMs. VISCON comes with a diverse set of probe image set to better investigate the factors of vision hallucinations, and comprehensive referential visual concept set as evaluation standard to detect vision hallucinations about objects, attributes and relations in model's reponse. VISCON employs both EMD-based and "Evaluate-By-Edit" based pipelines, providing a comprehensive analysis that blends quantitative assessment with interpretive clarity to understand hallucination in LVLMs. VISCON evaluates LVLMs through their captioning responses, offering better model discrimination by requiring them to handle multiple concepts simultaneously. Our empirical findings reveal key insights into the nature of vision hallucinations in LVLMs. We discovered that image domain shifts consistently lead to increased hallucinations, and more complex visual concepts such as relationships and attributes are particularly prone to hallucination. Additionally, we observed that longer model responses do not necessarily equate to more informative content and can, in fact, exacerbate the issue of hallucinations. In conclusion, VISCON represents a step forward in the evaluation and understanding of LVLMs, assessing the prevalent issue of vision hallucinations. As LVLMs advance and their applications grow in complexity and diversity, the methodologies and insights from our research could be beneficial in enhancing their applicability.

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

## A    APPENDIX OVERVIEW

In this supplementary appendix, we analyze additional LVLMs using extended evaluation pipelines and delve deeper into the methodologies and experiments underlying our VISCON metric.

First, in Appendix B, we extend the experiments conducted using VISCON. In Appendix B.1, we further analyze more factors of visual hallucinations, including the generation method of model response, and the model size. In Appendix B.2, we further analyze VISCON's stability and robustness against different prompts and LLMs used for concept extraction from model responses. Second, in Appendix C, we include conceptual visualizations related to the EMD-based evaluation (Appendix C.1), Evaluate-By-Edit evaluation (Appendix C.2) and visual concept reference set in VISCON probe image dataset (Appendix C.3), offering interpretable insights into the metrics we have proposed. Third, in Appendix D, we offer an in-depth exploration of VISCON's evaluation methodology. This includes detailed information on the data curation process for the probe image set in Appendix D.1, an in-depth look into the EMD-based evaluation pipeline in Appendix D.2, and an detailed explanation of the Evaluate-By-Edit pipeline in Appendix D.3. In Appendix D.4, we explained the QA-based evaluation metric baseline, combining it with our comprehensive reference visual concept set in VISCON. This section details our methodology for assessing visual hallucinations through question-answering. Each of these components is crucial to the robustness and efficacy of VISCON, and we provide insights into their design and functionality. Lastly, we discuss about the limitations and social impact of VISCON.

## B    ADDITIONAL QUANTITATIVE EXPERIMENTS

### B.1    MORE ANALYSIS OF HALLUCINATIONS

Table 6: EMD vs. generation method. Best and second best performances are marked **bold** and underlined. We conduct analysis with Phi-3V, and observe that beam search perform slightly better than nucleus sampling or greedy search.

| Generation Method | EMD↓ | | | |
| --- | --- | --- | --- | --- |
| | object | attribute | relation | all |
| Nucleus Sampling | 28.69 | 40.85 | 35.05 | 104.59 |
| Greedy | 28.30 | 40.83 | 34.95 | 104.09 |
| Beam Search | **28.23** | **40.73** | **34.88** | **103.85** |

**Impact of Generation Method** We analyze the relationship between hallucination metrics and three commonly used response generation methods for LVLMs: nucleus sampling (with $\text{temperature} = 0.9, \text{top-}p = 0.9, \text{top-}k = 50$), greedy search, and beam search (with beam size $= 5$). The experiments were conducted on the Phi-3V model, the best-performing model aside from GPT-4V, as the generation method for GPT-4V is not publicly available.

As depicted in Table 6, we found that beam search, despite its higher computational cost for inference, achieved the best performance in reducing hallucinations. This is likely due to beam search's better consideration of whole-sequence optimality by evaluating multiple possible sequences and selects the most probable one, rather than the token-wise optimality seen in greedy search. On the other hand, nucleus sampling performed the worst, potentially because its nature to generate more diverse responses by selecting suboptimal words in likelihood, that can lead to a loss in precision, increasing the risk of hallucinations.

Table 7: EMD vs. model size. Best performances are marked **bold**. We conduct analysis with LLaVA-1.5 (7B and 13B), and observe that larger models perform better on real-world similar images but worse on images with larger domain shifts.

| Model | EMD↓ | | | | | | | | | | |
|---|---|---|---|---|---|---|---|---|---|---|---|
| | mean | real-world | | | | | 3D | | | | |
| | | original | cartoon | sketch | oil painting | line | original | cartoon | sketch | oil painting | line |
| LLaVA-1.5 7B | **104.8** | 102.6 | 103.6 | 104.5 | 104.2 | **107.8** | **103.6** | 104.8 | **105.0** | **105.4** | **106.2** |
| LLaVA-1.5 13B | **104.8** | **102.4** | **102.7** | **103.1** | **104.1** | 109.0 | 104.0 | **104.4** | 105.5 | 105.9 | 107.0 |

**Impact of Model Size** We analyze the impact of model size on visual hallucinations by comparing different sizes of the same LVLM trained with identical data, training paradigms, and architecture. The experiments were conducted on the LLaVA-1.5 model, which has two published versions of different sizes (7B and 13B parameters), making it ideal for this analysis. Additionally, we compared the EMD-based metrics across different image domains, as depicted in Table 7.

We observe that larger model, despite its higher computational cost and parameter size, achieved only comparable performance in reducing hallucinations. The larger model performed better in real-world image domains but worse on images with significant domain shifts (e.g., line paintings, 3D rendered views). This suggests that for LLaVA-1.5, larger models might overfit to specific training image domains, increasing the likelihood of hallucinations when encountering unfamiliar images. This result showcase a evident trade-off between model size and generalization capability for LVLMs: larger models excel in familiar contexts but struggle with novel inputs, highlighting the need for balanced model training approaches to mitigate overfitting while maintaining high performance.

## B.2 More Analysis of VISCON

**EMD Stablity Against Prompts** We investigated different prompts with varying "length-control" words to control output detailedness (Table 8). Prompts like "in great detail" likely result in longer responses, while "concisely" may lead to shorter ones. Interestingly, GPT-4V can be controlled with exact word count prompts like "in 80 words," achieving an average response length close to 80. Comparing EMD scores and average response lengths, we found that when response length is within a mild range of 100±50 words, EMD scores only vary slightly (within ±0.5) across different length-control words. However, excessively long or short responses (>200 or <20 words) can increase hallucinations. Thus, EMD are quite robust against the variation of prompts leading to moderate lengths (e.g., 100±50 words).

**Stability Against Visual Concept Extraction Model** we conducted an ablation study using alternative LLMs for visual concept extraction from model response, using `claude-3.5-sonnet` and `LLaMA-3.1-70B` in addition to GPT-4 (`gpt-3.5-turbo-0613`) which is used in other experiments. Results in Table 9 and Table 10 show: 1) EMD metrics remain largely consistent across different LLMs 2) there are high correlations of EMD scores (Pearson r>0.96) when using different LLMs for visual concept extraction. These findings suggest that our concept extraction is

Table 8: Ablation of EMD against different length-control prompts. Prompts leading to too long ($\geq$ 200) or too short responses ($\leq$ 20) tends to induce more hallucinations.

| Model | Length-Control Prompts | Average Response Length (mean $\pm$ std) | EMD |
|---|---|---|---|
| GPT-4V | Describe the image in detail. | 227.78 $\pm$ 56.48 | 104.3 |
| GPT-4V | Describe this image in 80 words. | 85.82 $\pm$ 5.99 | 102.5 |
| Phi-3V | Describe the image in great detail. | 141.01 $\pm$ 49.85 | 104.8 |
| Phi-3V | Describe the image in detail. | 86.07 $\pm$ 21.05 | 104.6 |
| Phi-3V | Describe the image concisely. | 53.09 $\pm$ 16.19 | 105.2 |
| LLaVA-1.5 | Describe the image in detail. | 95.29 $\pm$ 14.91 | 104.8 |
| LLaVA-1.5 | Describe the image concisely. | 74.21 $\pm$ 17.28 | 104.6 |
| Qwen-VL | Describe the image in great detail. | 83.22 $\pm$ 39.23 | 105.3 |
| Qwen-VL | Describe the image in detail. | 84.15 $\pm$ 42.60 | 105.6 |
| Qwen-VL | Describe the image concisely. | 17.09 $\pm$ 13.84 | 108.4 |

robust across different LLMs and that the use of GPT-4 for concept extraction does not introduce significant bias in our evaluation framework.

Table 10: EMD metric stability against different LLM used for visual concept extraction.

| LLM for concept extraction | Evaluated LVLM | | | | | |
|---|---|---|---|---|---|---|
| | GPT-4V | Phi-3V | LLaVA-1.5 | Qwen-VL | InstructBLIP | Fuyu |
| GPT-4 (gpt-3.5-turbo-0613) | 104.3 | 104.6 | 104.8 | 105.6 | 106.1 | 107.0 |
| Claude-3.5 (claude-3.5-sonnet) | 103.0 | 103.7 | 104.3 | 104.7 | 105.9 | 106.5 |
| LLaMA-3.1 (LLaMA-3.1-70B) | 104.3 | 105.3 | 105.1 | 105.7 | 106.4 | 107.5 |

**Data Suf-fi-cien-cy**

Table 9: Pearson correlation analysis of different LLMs used for visual concept extraction.

| LLM | GPT-4 | Claude-3.5 | LLaMA-3.1 |
|---|---|---|---|
| Claude-3.5 | 0.9748 | - | - |
| LLaMA-3.1 | 0.9745 | 0.9647 | - |

To demonstrate the suffciency of current probe image scale, we assess the EMD metric's stability to the scale of probe image set. Analysis shows stable EMD metrics after using 50% of samples (Figure 7), indicating current size sufficiently reflects LVLM hallucination trends. Scaling beyond current dataset size may offer more comprehensive evaluation but less cost-efficient.

## C ADDITIONAL QUALITATIVE EXPERIMENTS

### C.1 VISUALIZATION OF EMD-BASED EVALUATION

To gain deeper insights into the behavior of our EMD-based evaluation pipeline, we visualize the optimal transportation matrix and the concept-wise EMD of the visual concepts mentioned by the model, as depicted in Figure 8 alongside an example image.

The visualized optimal transportation matrix reveals how semantic similarities between the reference and predicted visual concept sets are captured. We observe high transport coefficients between strong semantic correlated concepts, such as "couch" versus "sofa". This visualization effectively demonstrates the pipeline's ability capture semantic similarity between reference and predicted visual concept sets, despite terminology differences.

On the other hand, by examining the concept-wise EMD, we observe that more accurately predicted visual concepts, such as objects ("floor," "doorway," "couch," "window"), attributes ("desk is wooden", "window is glass"), and relations ("desk has objects"), correspond to lower EMD values. This correlation between prediction accuracy and EMD values empirically verifies the capability of our EMD-based evaluation that could distinguish hallucinations in visual concepts, allocating higher EMDs for more hallucinated predictions and lower EMDs for more accurate ones.

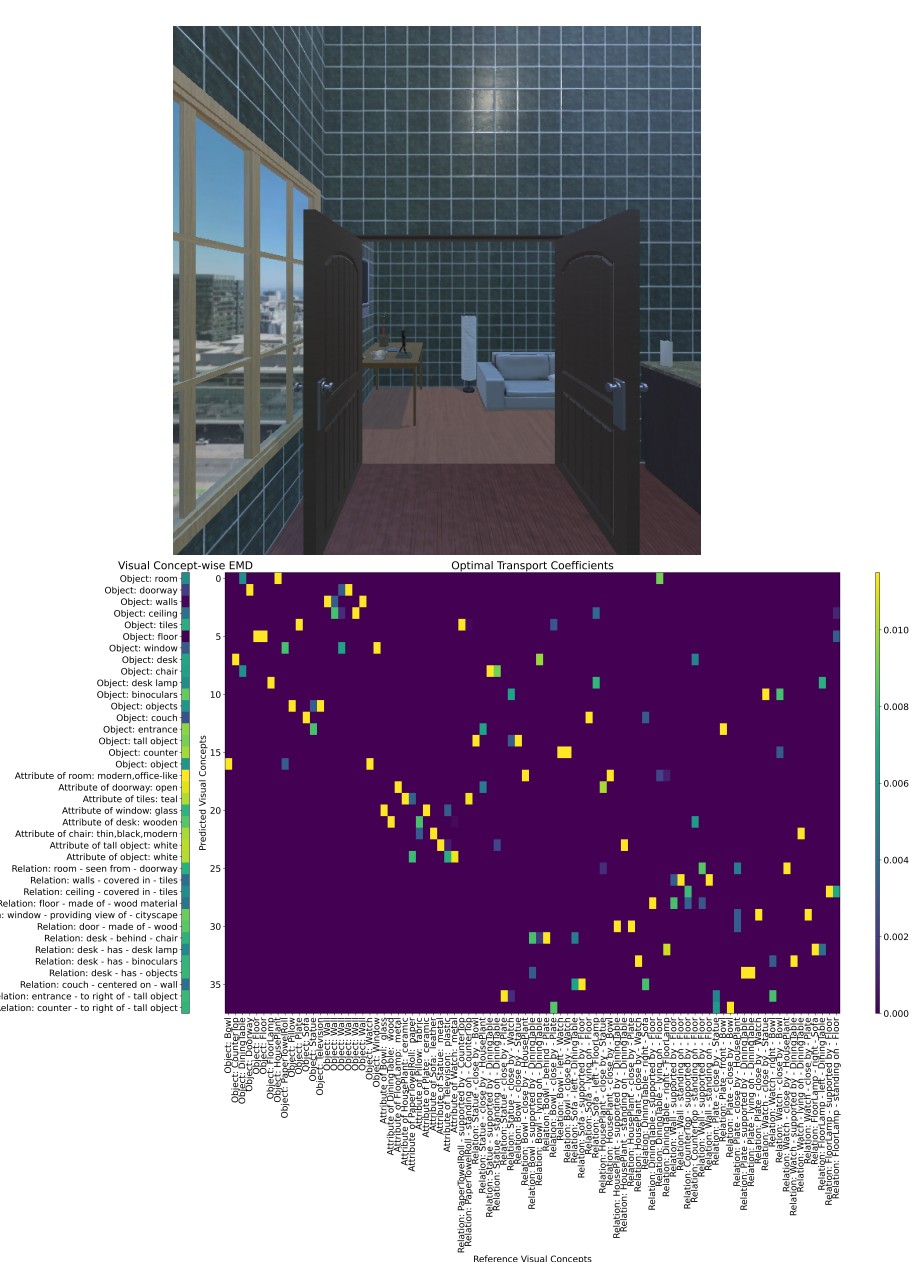

Figure 8: Visualization of concept-wise EMD distances and optimal transportation matrix for an exemplar image in EMD-based pipeline. This figure presents a detailed view of the Earth Mover's Distance (EMD) evaluation conducted on visual concepts as predicted by GPT-4V. Key aspects illustrated include the concept-wise EMD distances and the optimal transportation matrix, comparing the reference and the model-predicted visual concepts. The image shows that more accurate model predictions for various visual concepts, categorized as objects (e.g., floor, doorway, couch, window), attributes (e.g., 'desk is wooden', 'window is glass'), and relations (e.g., 'desk has objects'). Zoom in to observe the finer details of the EMD analysis and transportation matrix visualization.

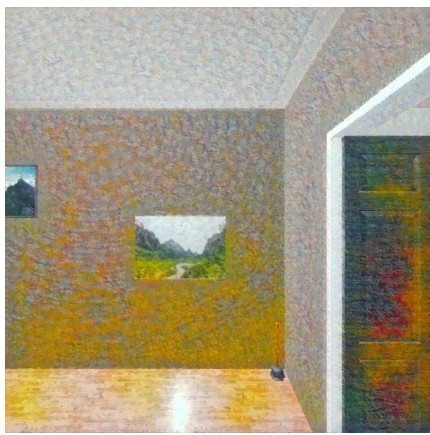

The image is a painting of a room with a wooden floor, featuring a large window that allows natural light to enter. The room has a slanted ceiling, and the walls are decorated with paintings made of paper and plastic. (The paintings are of various sizes) Some paintings appear (with some appearing) to be smaller and others larger. A cat is sitting on the floor near the window, enjoying the view. The room has a doorway, and there are two potted plants placed in the room, one near the cat and the other further away. A vase can also be seen, placed near the top right corner of the room.

Figure 9: Visualization of intermediate edit distance alignment during the query-and-revise procedure with an exemplar image in Evaluate-By-Edit pipeline. Red signifies words that have been either removed (not enclosed in parentheses) or substituted (enclosed in parentheses), and orange represents words that have been inserted as replacements. Consecutive edited words are re-organized together for clarity. It is observed that the majority of the modifications pertain to vision hallucinations, demonstrating the words where the LVLM generates incorrect outputs in an interpretable way.

These qualitative results from the visualization not only validate the effectiveness of the EMD-based evaluation in capturing semantic nuances but also highlight its potential in identifying and quantifying hallucinations in model predictions.

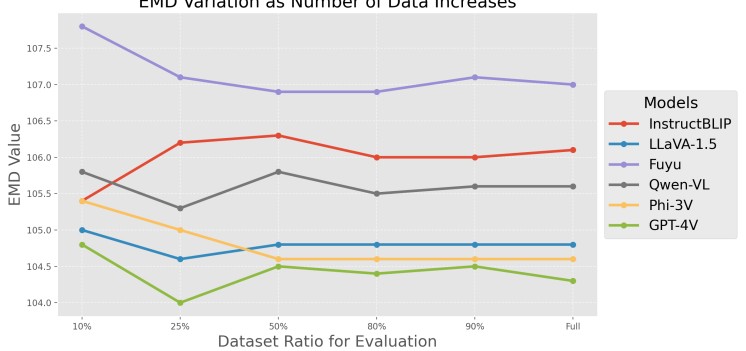

Figure 7: EMD metric vs. used data amount for evaluation.

## C.2 VISUALIZATION OF EVALUATE-BY-EDIT EVALUATION

To dig deeper for our Evaluate-By-Edit evaluation pipeline, we present a visualization of the edit process used in calculating the edit distance between the original model output and its revised counterpart. This process is depicted in Figure 9, alongside an example image. In the visualization, text modifications are highlighted with color to distinguish removed/substituted and inserted text. Although there are instances of erroneously removed text attributable to oversights of LLMs, the majority of the edits pertain to corrections of vision hallucinations. Specifically, these corrections address references to non-existent entities within the image, such as "a cat," "potted plant," and "a vase."

This qualitative result effectively demonstrates the pipeline's capability to validate each hallucinated visual concept. Moreover, it highlights the interpretability offered by the Evaluate-By-Edit pipeline, showcasing its utility in distinguishing and correcting vision hallucinations generated by LVLMs.

## C.3 VISUALIZATION OF VISUAL CONCEPT ANNOTATION IN VISCON

To illustrate the comprehensive nature of our visual concept annotations in the VISCON probe image dataset, we present a conceptual visualization in Figure 13. This figure showcases the richness and density of our annotation approach, which is crucial for effective hallucination evaluation and detection for LVLMs. Our method captures a wide array of visual elements within each image,

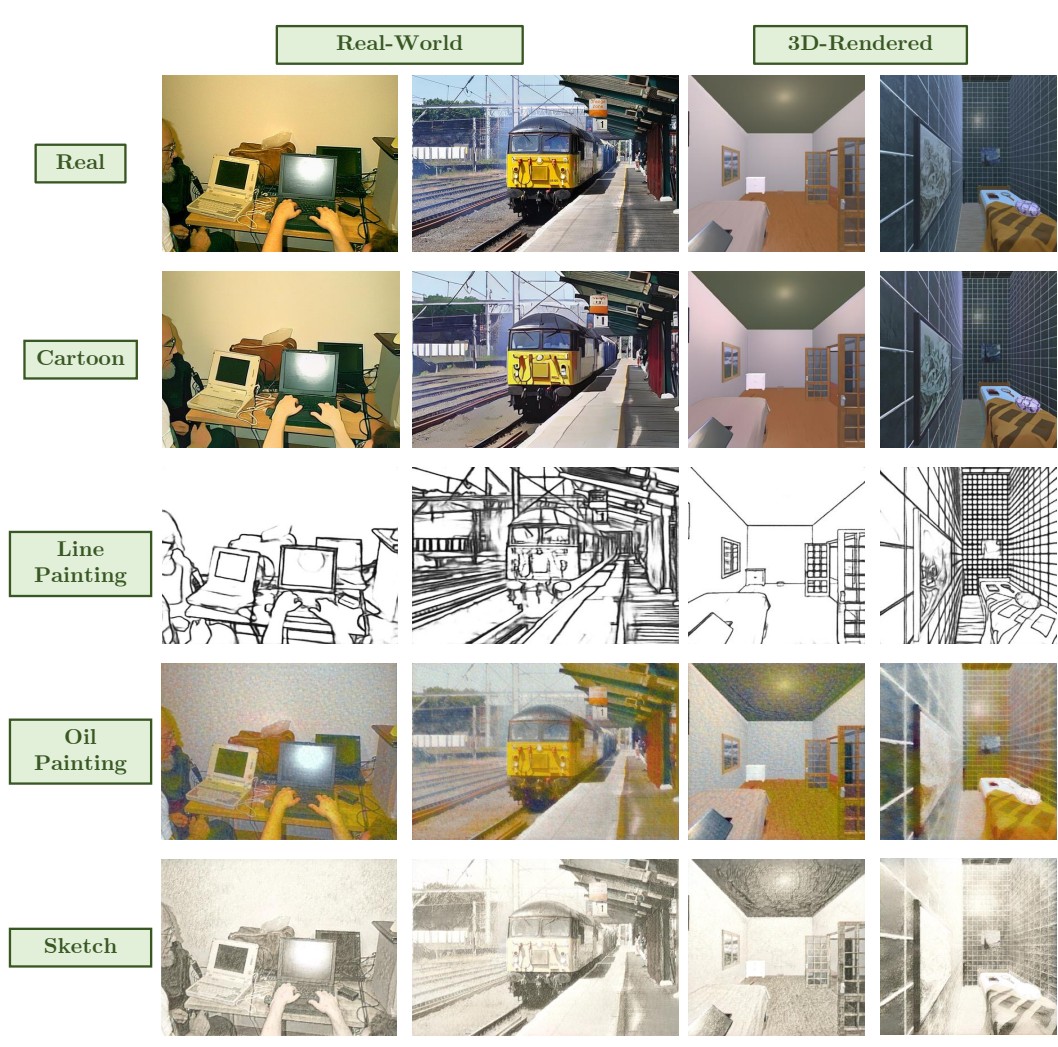

Figure 10: Exemplar Images from VISCON's Probe Image Set. Displayed are representative images from both real-world and 3D-rendered datasets, each stylized in four different ways. These variations serve to investigate the impact of visual domain shifts on LVLM hallucinations.

(a). object

[SYSTEM] You are a helpful assistant with advanced visual and linguistic context analysis function.
[USER] Extract the object entity words in appearance order in following sentence, ONLY output the entity words, ignore attributes or relations: an red apple under the blue desk
[ASSISTANT] apple, desk
{... MORE FEW-SHOT EXAMPLES...}
[USER] Extract the object entity words in appearance order in following sentence, ONLY output the entity words, ignore attributes or relations: {...model response...}

(b). attribute

[SYSTEM] You are a helpful assistant with advanced visual and linguistic context analysis function.
[USER] Identify the attributes of mentioned objects in the following sentence. If an object has multiple attributes, separate them with commas. If an object has no attributes, leave it blank after the colon. Ignore positional attributes relative to image such as 'at the center of image'. IGNORE contents of paintings or picutures, IGNORE the sentence incompleteness and errors. If there are reptitions in sentence, ignore repeated sentence part. Follow the above output format.; Sentence: "an antique clock with gold trimmings hanging on a stone wall"; Objects: clock,wall; Attributes:
[ASSISTANT] clock:antique, gold; wall:stone
{... MORE FEW-SHOT EXAMPLES...}
[USER] Identify the attributes of mentioned objects in the following sentence. If an object has multiple attributes, separate them with commas. If an object has no attributes, leave it blank after the colon. Ignore positional attributes relative to image such as 'at the center of image'. IGNORE contents of paintings or picutures, IGNORE the sentence incompleteness and errors. If there are reptitions in sentence, ignore repeated sentence part. Follow the above output format.
Sentence:; {...model response...}; Objects: {...extracted objects from (a) above...}; Attributes:

(c). relationship

[SYSTEM] You are a helpful assistant with advanced visual and linguistic context analysis function.
[USER] List the object relations in following sentence. Output each relation in 'object 1,predictate,object 2' format. Ignore positional relations about the image such as 'object, at the center, image'. IGNORE contents of paintings or picutures, IGNORE the sentence incompleteness or errors. If there are reptitions in sentence, ignore repeated sentence part. If no relation is found, output NONE. Follow the above output format.; Sentence: "A cat is sleeping on the rug in front of the fireplace"; Objects: cat, rag, fireplace; Relations:
[ASSISTANT] cat, on, rug;cat, in front of, fireplace
{... MORE FEW-SHOT EXAMPLES...}
[USER] List the object relations in following sentence. Output each relation in 'object 1,predictate,object 2' format. Ignore positional relations about the image such as 'object, at the center, image'. IGNORE contents of paintings or picutures, IGNORE the sentence incompleteness or errors. If there are reptitions in sentence, ignore repeated sentence part. If no relation is found, output NONE. Follow the above output format.; Sentence: {...model response...}; Objects: {...extracted objects from (a) above...}; Relations:

Figure 11: Used prompt for GPT-4 to automatically extract visual concepts from LVLM's response. "[SYSTEM]", "[USER]" and "[ASSISTANT]" are the role tags of each message of the prompt. **(a)**. object **(b)**. attribute **(c)**. relationship. The extracted object names from step **(a)** are used to inform later extraction of attributes and relationships. Some newlines are replaced with semicolons for presentation clarity.

[SYSTEM] You are an expert database assistant and programmer. Correct the following image caption labled after [Caption]. It is from a view of a 3D rendered indoor room. It may contain errors like wrong or not existing object, object relation or object attributes. Objects after [Seen Objects] label are all objects can be seen from the view, and are always correct.

[USER] You are an expert database assistant and programmer. Correct the following image caption labled after [Caption]. It is from a view of a 3D rendered indoor room. It may contain errors like wrong or not existing object, object relation or object attributes. Objects after [Seen Objects] label are all objects can be seen from the view, and are always correct.

—

Generate necessary JavaScript code-like API calls to access an database constructed with the image, including the objects, object relations and object attributes. According to the response/return value, correct the caption. If some object, relation or attribute does not exist, remove in the final caption.

—

[AVAILABLE API DOCUMENTATION]

$\langle$object X$\rangle$: in following documentation, this annotation represents an object with name X and an ID number indicating which object among same category. E.g., $plant_0$, $plant_1$, $dresser_0$, etc.

GetRelation($\langle$object A$\rangle$, $\langle$object B$\rangle$): return the relation of A to B. E.g., returns "supported by" or "close to".

GetDescription($\langle$object A$\rangle$): return the object description of one single object. E.g., return "an L-shaped blue sofa"

GetAttributes($\langle$object A$\rangle$): return a JSON list of the attributes that an object holds. E.g., return {"material": "plastic"}

—

If any API call corresponds to object, relation or attribute that does not exist, the API returns NONE. If the object, relation or attribute is not given in the database, the API returns KEEP. Objects with same name are identified by its numeral IDs. E.g.,$\langle$object $apple_1$$\rangle$ and $\langle$object $apple_2$$\rangle$ are two different apple objects.

—

$\langle$CASE {caption-id}$\rangle$

[Caption]: {caption}

[Seen Objects]: {objects}

—

First, generate the JavaScript API calls, each about single object, relation or attribute. ONLY output minimal necessary API calls, thus ignore the object, relation or attribute not involved in the caption. Begin API calls after output [BEGIN API]. Output [END API] after all calls. Responses will be given later, and stop after [END API]. [ASSISTANT]

{... LLM generate pseudo-code-like queries ...}

[USER] {... executed query results ...}

—

[Caption]: {caption}

—

Consider the API call return values of the room information. Correct the caption. REMOVE descriptions about non-existent objects, relations or attributes (those with NONE return values). KEEP the same descriptions in [Caption] about objects, relations or attributes with KEEP query result. ONLY output the caption.

—

[ASSISTANT]

{... LLM generate revised response ...}

Figure 12: Used prompt for GPT-4 in Evaluate-By-Edit-based pipeline, to automatically query visual concepts from database and revise LVLM's response. "[SYSTEM]", "[USER]" and "[ASSISTANT]" are the role tags of each message of the prompt.

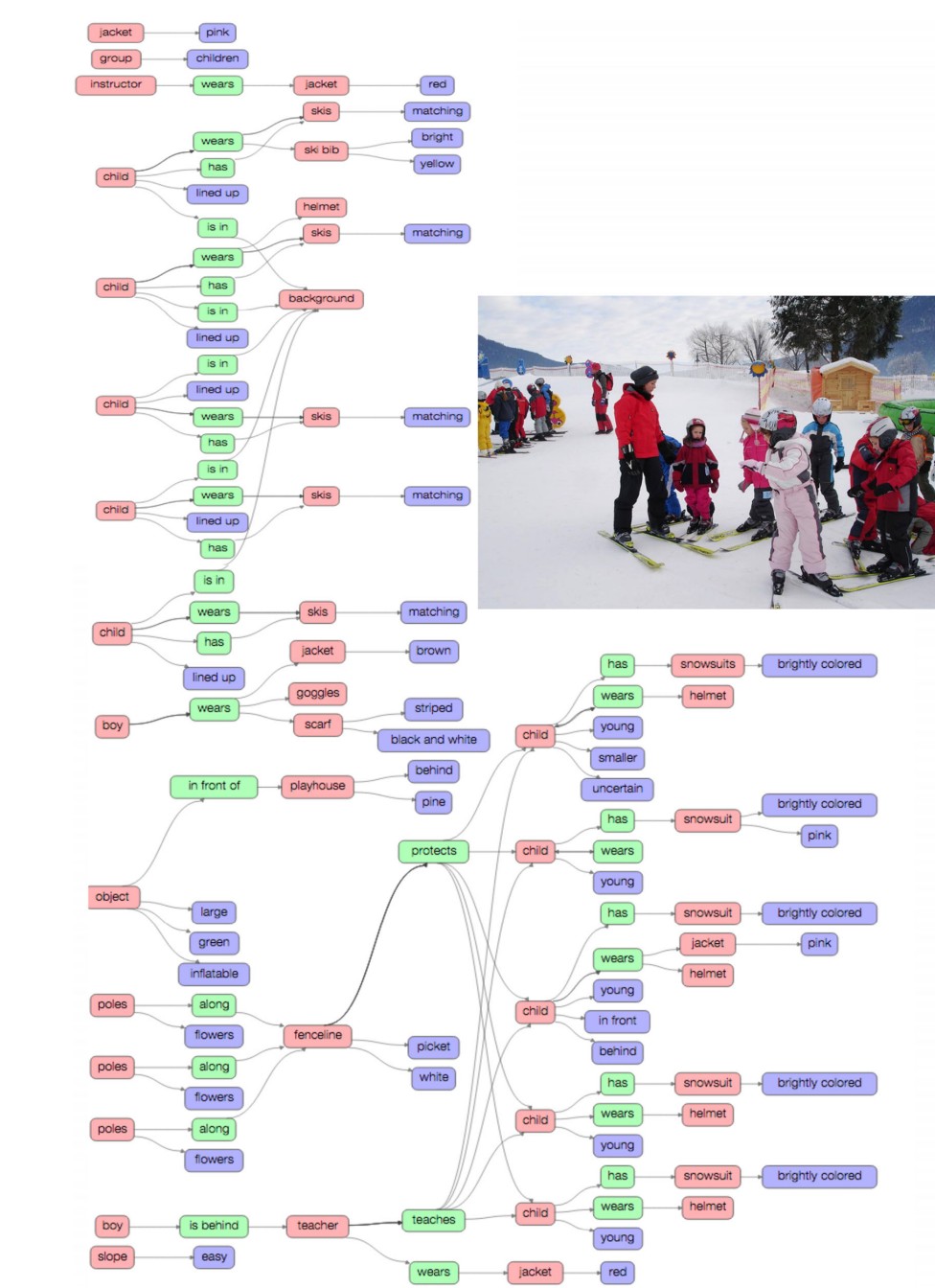

Figure 13: Exemplar VISCON visual concept reference set derived from image scene-graph. Rich visual concept annotations are provided to capture as dense as possible positive visual elements in image. Image reproduced from Visual Genome (Krishna et al., 2017), published under CC-BY license.

including objects, attributes, and relationships. This dense annotation strategy stands in contrast to previous benchmarks that often rely on sparse object labels. As depicted, the visual concept reference set is derived from the image's scene graph, allowing us to capture not just the presence of objects, but also their attributes and the relationships between them. By providing such detailed annotations, we aim to minimize false negatives in hallucination detection and offer a more nuanced evaluation of LVLM performance across complex visual scenes.

## D  DETAILS IN VISCON EVALUATION

In this section, we delve deeper into the methodologies employed in VISCON, providing a comprehensive understanding of our probe image selection, stylization processes, and the specifics of our Earth Mover's Distance (EMD)-based and Evaluate-By-Edit-based evaluation pipelines. In all experiments unless specifically mentioned, the model response is generated with prompt "Describe the image in detail". For GPT-4V, "Describe this image in 80 words" is used to control its response to a similar range with other LVLMs.

### D.1  DETAILED INSIGHTS INTO PROBE IMAGE SELECTION AND STYLIZATION TECHNIQUES

**Probe Image Set:** To evaluate LVLMs under various scenearios, we meticulously select a range of images with detailed visual concept annotations (Figure 10). We opt for the VisualGenome dataset (Krishna et al., 2017) for its rich scene-graph annotations, encompassing a wide array of visual concepts. Additionally, we generated views of 3D indoor scenes from the PROCTHOR dataset (Deitke et al., 2022), chosen specifically for its detailed metadata on visible objects, including names, attributes, colors, spatial positions, and bounding boxes.

In processing 3D-rendered images, we employ a careful selection criterion, discarding images with minimal object presence. Specifically, images capturing fewer than two objects were excluded, and we manually verify the images to be visually correct. For these images, we establish object relationships using a set of pre-defined rules. For instance, two objects were considered 'close to' each other if their distance fell within a certain threshold. We utilize 24 types of relation predicates inspired by the 3D-SSG dataset.

In the curation of our probe image set for VISCON, we aim for a balance between diversity and the practicality of resource utilization. While it is feasible to curate a large-scale image dataset automatically, the linear increase in LLM inference resources with the dataset's size posed a significant constraint. Therefore, we opt for a more moderate dataset size. Our selection process results in a diverse and representative set of images, comprising 46 real-world and 62 3D-rendered images without image stylizations, and 540 images if counting their stylizations as discussed below.

**Image Stylization Process:** To explore the influence of visual domain shifts on vision hallucinations, we apply stylization techniques to each image in our set. Four distinct styles are generated: sketch, line painting, oil painting, and cartoon. For the sketch and oil painting styles, we utilize the CMD method (Kalischek et al., 2021) for art style transfer, drawing inspiration from Claude Monet's "Autumn on the Seine, Argenteuil" for oil painting and Vincent van Gogh's "Village Street, Sketch, 1890" for sketch, to stylize the images accordingly. The cartoonization of images is achieved using the InstructPix2Pix model (Brooks et al., 2023)(Paul, 2023), while line painting effects were created via an edge detection model (Soria et al., 2020). In general, the image domain shift increase from cartoon and oil painting styles to sketch and line painting styles. Note that for 3D rendered views, cartoonized images are quite similar with the original image, due to the already flat color configuration in 3D rendered views. A visual representation of these stylizations is presented in Figure 10, and after the stylization process, 540 images are acquired for VISCON evaluation.

### D.2  MORE DETAILS IN EMD-BASED EVALUATION

In our EMD-based evaluation, we employ GPT-4 (`gpt-3.5-turbo-1106` version)(OpenAI, 2023) to extract visual concepts from the models' responses. The extraction process, detailed in Figure 11, involves specifically designed prompts for isolating objects, attributes, and relationships.

| Visual Concept Type | Template | Example |
|---|---|---|
| Object | Object: {object} | Object: wall |
| Attribute | Attribute of {object}: {attribute} | Attribute of wall: beige |
| Relationship | Relation: {object1} - {rel} - {object2} | Relation: framed picture - hangs on - wall |

Table 11: Used textual prompts for formatting visual concept for textual embedding and EMD calculation. Filled texts are marked as red.

To enhance the accuracy and stability of visual concept detection, we include a few hand-selected few-shot examples within these prompts.

The critical step of calculating the EMD involves comparing the visual concepts mentioned in the LVLM's responses (identified by GPT-4) with our reference set from VISCON. To facilitate this comparison, we format the visual concepts from the LVLM responses and the reference concepts using predefined templates (details and examples are provided in Table 11). The formatted texts are then processed through a state-of-the-art sentence-level embedding model, namely GTE (Li et al., 2023c), to obtain their sentence embeddings. This ensures that each visual concept, whether from the LVLM's mentions or the reference set, is embedded into the same linguistic semantics space. These embeddings are subsequently used to compute the EMD metrics. For a more granular analysis, we calculate EMD values separately for objects, attributes, and relationships, as illustrated in Table 3 of the main paper. This approach allows for a detailed evaluation of the LVLMs' performance across different types of visual information.

### D.3   MORE DETAILS IN EVALUATE-BY-EDIT-BASED EVALUATION

In our Evaluate-By-Edit approach, detailed in Section 3.4 of the main paper, we utilize GPT-4 for a two-step process: generating queries based on the LVLM's response and then revising this response accordingly. Figure 12 illustrates the prompts used for this query-and-revise procedure.

**Query Execution and Response Revision:** Upon executing a query, we encounter three potential outcomes: 1) **Non-existence Confirmation**: If the query confirms the non-existence of a visual concept (such as an object or relationship not present in the scene-graph), it returns "NONE." This result prompts GPT-4 to remove the corresponding descriptions from the LVLM's response. 2) **Uncertainty**: In cases where the existence of a concept (like an attribute or relation) is uncertain, possibly due to incomplete annotations, the query returns "KEEP." This instructs GPT-4 to leave the related text unchanged in the revised response. 3) **Existence Confirmation**: When the query confirms the existence of a visual concept, it returns the exact visual concept information found in the visual concept database. This result prompts GPT-4 to modify the original description in the model response to align with the more accurate information found in the visual concept database.

After receiving the outputs of each query, we further prompt GPT-4 to revise the original model response based on the query execution results. This involves either removing incorrect information, keeping the text unchanged, or modifying parts of the text to be more accurate. This process ensures that the revised responses are as accurate and reliable as possible.

**Evaluating Hallucinations through Edit Distance:** After acquiring the revised model response, our focus shifts to evaluating hallucinations by analyzing the word-level edit distance between the original and revised LVLM responses. This process begins by tokenizing the sentences into individual words, facilitating a detailed, word-level evaluation. Through this method, we can precisely quantify the extent of revisions needed, aligning the LVLMs' outputs with a more accurate visual representation of the images. This approach offers a granular insight into the nature and extent of hallucinations present in the LVLM's original responses.

**Limitations against Image Domain Variations:**  Empirically, we observe that edit distance metrics are significantly influenced by the stylistic variations in model outputs across visual domains, which may include additional image's style descriptions (e.g., "this image appears to be a digitally altered representation of ..."). Such stylistic texts are tend to be remained by the revise procedure, which impact intercept metrics, and introduce confounders into the comparison that are hard to mitigate due to the free-form nature of model responses. EMD-based pipeline ignores these texts, thus are robust to model response styles. Therefore, our analysis primarily focuses on examining the impact of domain gaps on EMD evaluations.

### D.4   MORE DETAILES IN QUESTION-ANSWERING (QA) BASED EVALUATION

In this section, we broaden our evaluation pipelines of VISCON with a question-answering (QA) based metric inspired by POPE. POPE asks LVLMs about the existence of all objects in an image, as well as a negative set of non-existent objects, to evaluate misrecognitions of image contents. We propose extending this to ask about the existence of objects, attributes, and relations, using

a comprehensive set of annotated visual concepts from scene-graphs. This QA-based evaluation provides a fuller assessment of visual hallucinations compared to POPE, which only queries objects.

Specifically, in our QA-based evaluation pipeline, we ask the LVLMs about each annotated object, attribute, and relation. For example, questions can be: "Is there a desk?" (object), "Is the apple red?" (attribute), or "Is the cabinet beside the TV?" (relation). We generate a set of negative questions by randomly replacing one element of a visual concept with a non-existent one. For objects, the object name is replaced by a randomly selected non-existent one; for attributes, either the object name or attribute name is replaced; and for relations, the object, subject name, or relation predicate is replaced. We query the LVLMs with the generated set of positive and negative questions and compute QA accuracy as a metric of visual hallucinations, and we collect a total of 100K questions from the probe image set in VISCON.

## E  LIMITATIONS AND SOCIAL IMPACT

**Limitations** Despite the comprehensive visual concepts and meticulously designed evaluation pipelines in VISCON, our method has some limitations: 1) **Influence of Stylistic Variations on Edit Distance Metric:** Edit distance metrics are significantly affected by stylistic variations in model outputs across visual domains. These variations often include additional style descriptions of the image (e.g., "this image appears to be a digitally altered representation of ..."). Such stylistic texts remain after the revision process, impacting intercept metrics and introducing confounders that are difficult to mitigate due to the free-form nature of model responses. Therefore, our analysis primarily focuses on examining the impact of domain gaps on EMD-based evaluations. 2) **Coverage of Probe Image Set**: Our probe image set does not encompass all image types that are truly "unseen" by current LVLMs. Certain types of hallucinations might not be evaluated and showcased by our metrics. For instance, highly abstract or conceptually complex images, such as avant-garde art or specialized medical imaging, may not be adequately represented in our current dataset. This limitation suggests that there could be other hallucination patterns that remain unexplored.

**Social Impact** The evaluation of visual hallucinations in LVLMs is crucial for enhancing the safety and reliability of these AI systems. As LVLMs are integrated into various applications such as intelligent assistants, virtual reality experiences, and educational tools, ensuring their accuracy and trustworthiness becomes paramount. Our rigorous evaluation methodology helps identify and mitigate instances of hallucinated or erroneous responses. Accurate visual and language processing is essential in contexts where incorrect information could lead to harmful consequences, such as healthcare or education. By asseding LVLM hallucinations, our method could help enhance their reliability and user trust, making it essential for fostering public trust and ensuring the ethical use of AI systems in diverse applications.

