# OpenReview forum: "VISCON: Identifying and Benchmarking Vision Hallucination for Large Vision-Language Model"
_ICLR.cc/2025/Conference — Submitted to ICLR 2025_

### Official Review · Reviewer_kDMt · 2024-10-30

**Soundness:** 3
**Presentation:** 4
**Contribution:** 3
**Rating:** 5
**Confidence:** 4

**Summary:**

This paper introduces a novel framework, VISCON, which comprises a benchmark dataset and quantitative evaluation pipelines designed to detect hallucinations in the outputs of existing Large Vision-Language Models (LVLMs). Compared to previous benchmarks, such as POPE, VISCON encompasses a wider variety of image styles across multiple visual domains and a broader range of visual concepts. Additionally, it proposes two innovative evaluation pipelines, namely EMD and Evaluate-By-Edit. Extensive experiments validate the efficacy and robustness of VISCON in assessing the hallucination potential of LVLMs.

**Strengths:**

1. The motivation behind this study is clear, and the method for dataset collection and evaluation is intuitive.
2. Compared to previous benchmarks, VISCON encompasses a broader range of visual concepts, such as object relations, enabling a more comprehensive evaluation. This is crucial, as prior works often generate numerous false negative samples.
3. The experiments are exhaustive and easy to follow, effectively verifying the robustness of this type of hallucination evaluation.
4. Overall, the paper is well-written and easy to read.

**Weaknesses:**

1. The evaluation pipeline is quite complex and tedious. For instance, in the case of EMD, the process begins with extracting visual concepts from the detailed caption using GPT-4. Following this, it is necessary to obtain the text embedding for each visual concept before the final evaluation. I am concerned that such a long may hinder its adoption within the community.
2. This type of evaluation can be easily manipulated. For example, if a model is trained to generate short captions, even when given the prompt, "describe the image in detail," it may hallucinate less than a model that generates longer captions. Consequently, the performance of such a model might be overestimated.
3. I am unable to find detailed statistics about the evaluation dataset constructed by this paper. Additionally, there is a lack of information regarding the detailed procedure for constructing the evaluation dataset, including aspects like annotation and sanity checks.

**Questions:**

See weakness

---

> ### Author Response · Authors · 2024-11-30
> **Response to Reviewer kDMt**
>
> We sincerely thank Reviewer kDMt for their thorough evaluation of our work and appreciate the opportunity to further clarify important aspects of our research. We apologize for the delay in our response and hope that our revisions address your concerns. Please feel free to reach out for further clarification or justification if needed.
>
> **W1**. EMD calculation is complex to apply.
>
> Briefly speaking, EMD-based evaluation is more complex than using LLMs end-to-end or asking concept-wise questions. However, it evaluates LVLMs better in real-world application scenarios in a more reliable way.
>
> In previou methods, there are several straightforward method to evaluate hallucinations based on visual concept annotations.
> While using LLMs/VLMs to directly evaluate model response (e.g., FaithScore) against reference visual concepts might seem straightforward and simple for evaluating hallucinations, they have notable limitations. LLMs/VLMs themselves can introduce subjective errors in such complex task handling long caption and multiple visual concepts together, and often lack consistency across different samples, making it hard to distinguish between major and minor hallucinations (e.g., confusing "cat" with "dog" versus "computer monitor" with "laptop"). As a result, FaithScore only achieved a Pearson correlation of 0.48 while VISON achieved a much higher value of 0.95. Additionally, discriminative (i.e., QA-based) metrics (e.g., POPE, AMBER) that simply query LVLMs about the existence of specific visual concepts are too simplistic for real-world applications, where models need to handle multiple concepts simultaneously, such as in detailed captions or multi-turn questions.
> In contrast, despite more complex, our EMD-based approach offers several key advantages to address these deficiencies of current benchmarks: 1) we only abstract visual concepts with LLM, reducing ambiguity and simplifying the task for LLMs, which only require LLMs to repeat specific words from the input. 2) we use specialized sentence embedding models that are better suited for discriminating text pairs, providing nuanced, soft similarity scores rather than binary 0/1 values. 3) we evaluate LVLM's hallucinations through their long response when required to describe image in details, better discriminating different models with this more complex task.
> By delegating the evaluation to two simpler tasks and specialized models—concept extraction and similarity discrimination—we enhance reliability and ensure intra-sample consistency, which end-to-end LLM-based scores often lack. By evaluating through detailed captions, our metric better discriminate different LVLMs.
> Our benchmark's effectiveness is further validated by human preference alignment experiments, as shown in Table 5, demonstrating its superiority over previous methods and simpler QA-based evaluations.
>
>
> **W2**. If model tends to respond shorter, it may lead to less hallucination evaluation result.
>
> We appreciate the reviewer's observation that the tendency of models to produce shorter captions, may lead to lower hallucination evaluation scores. However, it is important to clarify that our metrics are specifically designed to assess hallucination in actual model outputs during captioning tasks. While shorter captions may yield less hallucinations, this phenomenon reflects a fundamental trade-off between providing informative descriptions (requiring longer captions) and maintaining factual accuracy (easier with shorter captions), rather than a limitation of our evaluation method.
>
> Moreover, to mitigate the impact of caption length on hallucination measurement results in VISCON, we have implemented two key strategies:
>     - Output Length Control: We observe that most current LVLMs naturally produce outputs within a moderate length range (100 ± 50 words), with the exception of GPT-4V, which tends to generate longer responses. To address this, we control GPT-4V's output length by including specific instructions in the prompt, such as "in 80 words," as detailed in Appendix B.2. This approach helps standardize output lengths and ensures consistent evaluation across models. For future models with more variable output lengths, similar prompt engineering techniques can be applied to maintain consistency.
>     - Normalization in Edit-Distance Metric: The Edit-Distance metric further accounts for caption length by normalizing the total edit distance by the caption length. This normalization ensures that the metric remains robust to variations in caption length, allowing for fair comparisons across models, even when caption lengths vary slightly.
>
> These measures ensure that our evaluation framework remains fair and effective against different detailedness level tendency of models.

---

> > ### Author Response · Authors · 2024-11-30
> > **Response to Reviewer kDMt (Continued)**
> >
> > **W3**. Unclear data statistics, construction method, annotation procedure and sanity checks.
> >
> > We have provided a comprehensive overview of our data construction method and evaluation procedure in Appendix D. Our dataset includes 540 images, encompassing a wide range of visual concepts with a total of 19K object, 18K attribute, and 37K relation annotations. This significantly exceeds previous benchmarks in both the quantity and density of visual concepts. As for a sanity check, we included a scene-graph annotation example in Figure 13 of the appendix. This example demonstrates the level of detail and accuracy in our annotations.

---

### Official Review · Reviewer_58J2 · 2024-11-03

**Soundness:** 3
**Presentation:** 2
**Contribution:** 3
**Rating:** 5
**Confidence:** 4

**Summary:**

This paper focuses on the evaluation of hallucination in large vision-language models. It proposes an evaluation benchmark and an automatic pipeline to conduct quantitative evaluation. The benchmark comprises of real world images and 3D rendered images, which are further augmented by image stylization. The benchmark includes two modes: an Earth Mover’s Distance and an Edit Distance. The evaluation reveals some interesting phenomenon in LVLMs.

**Strengths:**

1. The benchmark is curated and annotated comprehensively by considering various image domains, styles, and dense annotations (concepts).
2. The evaluation pipeline provides new perspectives to this venue.
3. The experiments and analysis are sufficient.

**Weaknesses:**

1. There are more related works that haven’t been discussed and compared, especially those that are also using Visual Genome and are also considering more concepts beyond object categories, e.g., Table 1 in [1].
2. Visual Genome is a widely used dataset to curate benchmark. In this work, the authors involve 3D rendered images into this benchmark. However, the rendered images are all constrained in indoor room scene. This would inevitably restrict the domain of the data, which actually somehow contradict with the main claim of the paper.
3. Since the visual concepts are come from the scene-graph annotation from Visual Genome, as stated in line 101, how do the authors ensure their generated annotation is more comprehensive than other works that also use visual genome scene-graph? Although there are some comparison for this aspect, it is not clear how does it achieved.
4. I wonder would the evaluation framework, especially the Edit Distance version, prefers short captions? For example, the model can be very lazy, simply describing the main objects to avoid making mistakes. Is the framework robust to such cases? I noticed that there are some discussions regarding caption lengths in Figure 5, but it appears to be hard to read without sufficient explanations. It's better to investigate this issue in detail.
5. The EMD is based on an embedding space of a text encoder. However, as discussed by previous works[1], some text encoders, e.g., CLIP text encoder, has significant bias to object categories, while lacking capabilities on telling attributes and relations. Besides, for most text encoders, distinguishing similar visual concepts with small difference is a very challenging task, while such kind of cases are very typical in visual hallucination. The authors should better cover more implementation details and investigations on this aspect, as it may fundamentally invalidate the assumption of EMD.
    - How does the text encoder perform on attributes and relations compared to objects?
    - How well does it distinguish between similar visual concepts?
    - What investigations have you done to ensure the validity of the EMD assumption given these potential limitations?
6. Similarly, in Edit Distance, how to ensure the refined captions are reliable?
7. In general, the method, including EMD and Edit Distance, seems to be proposed out of thin air without enough motivation. Could you give some motivation about the design? Why do you think it is good?

[1] Hallucination of Multimodal Large Language Models: A Survey. arXiv 2024.
[2] When and why vision-language models behave like bags-of-words, and what to do about it? ICLR 2023.

**Questions:**

Refer to weaknesses.

---

> ### Author Response · Authors · 2024-11-30
> **Response to Reviewer 58J2**
>
> We sincerely thank Reviewer 58J2 for their thorough evaluation of our work and appreciate the opportunity to further clarify important aspects of our research. We apologize for the delay in our response and hope that our revisions address your concerns. Please feel free to reach out for further clarification or justification if needed.
>
> **W1, W3**. Need discussion on the differences of previous work (e.g., AMBER, PhD, FaithScore, Reefknot). How VISCON's annotated visual concept more comprehensive than previous work also using VisualGenome (e.g., CCEval)?
>
> Quantitatively comparing our method to previous approaches is challenging due to the diversity of datasets and benchmarks, many of which are still in preprint form and not yet published (e.g., AMBER, PhD, FaithScore, Reefknot, CCEval, as reviewers indicate). According to the ICLR 2025 Reviewer Guide, it is not mandatory to discuss and compare with unpublished or very recent works. However, we can highlight key differences between our approach and several representative existing methods:
> - *FaithScore*: This method relies on visual entailment models, which are themselves are VLMs and prone to hallucinations, compromising their reliability. In contrast, our method uses dense human annotations to directly evaluate hallucinations, providing a more reliable and comprehensive assessment.
> - *CCEval*: This method extends reference object annotations using VisualGenome but still focuses primarily on objects and real-world photos like POPE, limiting its ability to capture the full range of visual concepts, such as attributes and relationships. Our method evaluates hallucinations across three types of visual concepts and multiple image domains, offering a more comprehensive evaluation of LVLM performance and investigation on the causes of hallucinations.
> - *AMBER, PhD and Reefknot*: These methods extend hallucination benchmarks to include attributes and relationships but face limitations. They either struggle with real-world scenarios involving multiple visual concepts due to their discriminative evaluation methods (only involving one concept per question) or lack dense annotations, leading to false negatives. Our method addresses these deficiencies by evaluating detailed caption responses, better discriminating LVLMs in complex tasks involving handling multiple concepts together, and enriching visual concept annotations with dense data from VisualGenome and oracle annotations from a 3D simulator, significantly reducing false negatives and enhancing reliability.
>
> Beyond the extensive visual concept types and annotation density that VISCON provides, our method uniquely explores the impact of image domains on LVLM hallucinations, a factor not addressed in previous research. Moreover, our experiments show that our method aligns more closely with human judgments, as demonstrated in the updated Table 5, highlighting its effectiveness and reliability in real-world scenarios.
>
> Regarding the visual concept annotation comprehensiveness and density, our method surpass previous method like CCEval by 1) considering more visual concept types, rather than only focusing objects in CCEval, 2) considering another series of image domain from 3D simulator, that also enrich the comprehensiveness of VISCON.
>
>
> **W2**. Rendered images are limited indoor scene.
>
>
> While our current dataset includes a significant number of outdoor images sourced from VisualGenome, we focused on 3D indoor scenes primarily because they are readily available from existing 3D simulators. This choice allows us to leverage detailed scene-graph annotations, which are crucial for our evaluation framework. However, we recognize the importance of expanding the diversity of image domains. In the future, we plan to incorporate more outdoor images, potentially from realistic environments in games like GTA V, to enhance the robustness and applicability of our benchmark across different settings.

---

> > ### Author Response · Authors · 2024-11-30
> > **Response to Reviewer 58J2 (Continued)**
> >
> > **W4**. If model tends to respond shorter, it may lead to less hallucination evaluation result.
> >
> >   We appreciate the reviewer's observation that the tendency of models to produce shorter captions, may lead to lower hallucination evaluation scores. However, it is important to clarify that our metrics are specifically designed to assess hallucination in actual model outputs during captioning tasks. While shorter captions may yield less hallucinations, this phenomenon reflects a fundamental trade-off between providing informative descriptions (requiring longer captions) and maintaining factual accuracy (easier with shorter captions), rather than a limitation of our evaluation method.
> >
> > Moreover, to mitigate the impact of caption length on hallucination measurement results in VISCON, we have implemented two key strategies:
> >     - Output Length Control: We observe that most current LVLMs naturally produce outputs within a moderate length range (100 ± 50 words), with the exception of GPT-4V, which tends to generate longer responses. To address this, we control GPT-4V's output length by including specific instructions in the prompt, such as "in 80 words," as detailed in Appendix B.2. This approach helps standardize output lengths and ensures consistent evaluation across models. For future models with more variable output lengths, similar prompt engineering techniques can be applied to maintain consistency.
> >     - Normalization in Edit-Distance Metric: The Edit-Distance metric further accounts for caption length by normalizing the total edit distance by the caption length. This normalization ensures that the metric remains robust to variations in caption length, allowing for fair comparisons across models, even when caption lengths vary slightly.
> >
> > These measures ensure that our evaluation framework remains fair and effective against different detailedness level tendency of models.
> >
> > **W5**. Text encoders like CLIP may ignore attribute/relation instead of object, which is key to hallucination evaluation.
> >
> > To clarify, in VISCON, we chose to use text-only sentence embeddings rather than CLIP-like multimodal models, which is deliberate and beneficial because: 1) text-only embeddings are trained on extensive text corpora, which better capture attributes and relations compared to CLIP's focus on captions, 2) these embeddings are designed to recognize semantic similarities and differences between text pairs, making them adept at distinguishing subtle concepts in language, 3) unlike multimodal models like CLIP, text-only embeddings do not suffer from visual biases. For instance, two visually similar objects might have similar textual embedding in CLIP due to vision alignment, despite large semantic differences, but text embeddings from text-only models can differentiate since there is no such vision alignment procedure.
> >
> > To demonstrate the effectiveness of text encoders, we conducted experiments comparing the similarities of attributes and relations. We evaluated pairwise similarities on three sets of objects (similar object, different object), attributes (same object, different attribute; same object, similar attribute) and relations (same object, different relation; same object, similar relation).  We observe that: similar concepts and different concepts have independent (validated by Welch's t-test) and large deviated similarities (validated by large effect size of Cohen's d > 0.8). The results, shown in the accompanying table, confirm that our adopted text encoders can effectively distinguish visual concepts on all three types of visual concepts, demonstrating its effectiveness in VISCON.
> >
> >
> > | Type      | Similar (mean ±std) | Different (mean ±std) | Independent? (p<0.01) | Effect Size (Cohen's d) |
> > |-----------|---------------------|-----------------------|-----------------------|-------------------------|
> > | Object    | 0.8804 (±0.0303)    | 0.7899 (±0.0241)      | ✓                     | 3.3                     |
> > | Attribute | 0.9201 (±0.0334)    | 0.8584 (±0.0525)      | ✓                     | 1.4                     |
> > | Relation  | 0.9619 (±0.0189)    | 0.8909 (±0.0377)      | ✓                     | 2.4                     |

---

> > > ### Author Response · Authors · 2024-11-30
> > > **Response to Reviewer 58J2 (Continued)**
> > >
> > > **W6**. Need to show caption refinement is reliable.
> > >
> > > To address the need for demonstrating the robustness of the edit distance in our "Evaluate-By-Edit" pipeline, we have included qualitative results in Figure 9 of the appendix. This figure provides a detailed visualization of the edit process used to calculate the edit distance between the original model output and its revised version. In this visualization, we highlight text modifications with different colors to clearly show which parts of the text were removed, substituted, or inserted.
> > >
> > > As shown in Figure 9, the majority of the edits effectively correct vision hallucinations. These corrections specifically target references to entities that do not exist in the image, such as "a cat," "potted plant," and "a vase." Inconclusive contents from annotations, such as "some paintings appear to be smaller" is remained as-is.
> > > This visualization demonstrates how our method accurately identifies and corrects hallucinations, reinforcing the robustness of the edit distance as a measure in our evaluation pipeline.
> > >
> > >
> > > **W7**. Unclear motivation of EMD and Edit Distance.
> > >
> > > In previou methods, there are several straightforward method to evaluate hallucinations based on visual concept annotations.
> > > While using LLMs/VLMs to directly evaluate model response (e.g., FaithScore) against reference visual concepts might seem straightforward and simple for evaluating hallucinations, they have notable limitations. LLMs/VLMs themselves can introduce subjective errors in such complex task handling long caption and multiple visual concepts together, and often lack consistency across different samples, making it hard to distinguish between major and minor hallucinations (e.g., confusing "cat" with "dog" versus "computer monitor" with "laptop"). As a result, FaithScore only achieved a Pearson correlation of 0.48 while VISON achieved a much higher value of 0.95. Additionally, discriminative (i.e., QA-based) metrics (e.g., POPE, AMBER) that simply query LVLMs about the existence of specific visual concepts are too simplistic for real-world applications, where models need to handle multiple concepts simultaneously, such as in detailed captions or multi-turn questions.
> > > In contrast, despite more complex, our EMD-based approach offers several key advantages to address these deficiencies of current benchmarks: 1) we only abstract visual concepts with LLM, reducing ambiguity and simplifying the task for LLMs, which only require LLMs to repeat specific words from the input. 2) we use specialized sentence embedding models that are better suited for discriminating text pairs, providing nuanced, soft similarity scores rather than binary 0/1 values. 3) we evaluate LVLM's hallucinations through their long response when required to describe image in details, better discriminating different models with this more complex task.
> > > By delegating the evaluation to two simpler tasks and specialized models—concept extraction and similarity discrimination—we enhance reliability and ensure intra-sample consistency, which end-to-end LLM-based scores often lack. By evaluating through detailed captions, our metric better discriminate different LVLMs.
> > > Our benchmark's effectiveness is further validated by human preference alignment experiments, as shown in Table 5, demonstrating its superiority over previous methods and simpler QA-based evaluations.
> > > On the other hand, our edit distance based pipeline aims to provide interpretability in hallucination evaluations by highlighting the specific changes needed to align model outputs with a refined version.

---

### Official Review · Reviewer_erAn · 2024-11-03

**Soundness:** 2
**Presentation:** 2
**Contribution:** 1
**Rating:** 3
**Confidence:** 4

**Summary:**

VISCON is a benchmark framework designed to evaluate vision hallucinations in Large Vision-Language Models (LVLMs) by assessing their consistency in generating text aligned with images. It includes a diverse dataset and two innovative pipelines: an Earth Mover’s Distance (EMD)-based method for distributional similarity and an "Evaluate-By-Edit" approach to gauge response alignment with annotated visual concepts. Extensive experiments on six LVLMs show that VISCON offers insights into hallucination factors, such as domain shifts and visual complexity, while providing results that better align with human evaluation than previous metrics.

**Strengths:**

1. **Comprehensive Evaluation Framework**: VISCON introduces a structured evaluation framework that includes diverse image styles and evaluates a broad range of visual concepts (objects, attributes, and relationships), enhancing its utility in analyzing vision hallucinations comprehensively.
2. **Innovative Metrics**: The Earth Mover's Distance-based pipeline and "Evaluate-By-Edit" method provide novel approaches to measure concept consistency, making VISCON more robust to vocabulary shifts and complex visual relationships, which are limitations in previous hallucination metrics.
3. **Alignment with Human Evaluation**: Extensive experiments reveal that VISCON aligns more closely with human preferences compared to established metrics, demonstrating its relevance and potential for real-world application.

**Weaknesses:**

1. **Limited Benchmark Validation**: The effectiveness of VISCON’s metrics has not been validated on a broader set of benchmarks, limiting confidence in its general applicability across various datasets.
2. **Lack of Clarity in Pre-defined Rules**: The framework lacks clarity around the pre-defined rules and dataset selection process, particularly in ensuring attribute consistency after stylization, which could impact the reliability of results.
3. **Insufficient Benchmark Comparison**: While VISCON emphasizes dense annotation, its benchmark is relatively small and its validation relies primarily on comparisons with POPE and MSCOCO, rather than a broader set of dense annotation benchmarks like Reefknot and it previous work, limiting the robustness of its claims.

**Questions:**

**Metric Validation**: Have VISCON’s metrics, Evaluate-By-EMD and Evaluate-By-Edit been tested on other dense annotation benchmarks?

**Benchmark Consistency**: What criteria were used to ensure the reliability and consistency of annotations across diverse visual domains? For example, how does VISCON handle potential changes in attributes after stylization, and could clarifying these processes enhance the benchmark’s reliability?

**Insufficient Benchmark Comparison and Size**: VISCON’s validation relies primarily on comparisons with POPE and MSCOCO, and its benchmark dataset is relatively small. Would expanding VISCON's benchmark and comparing it with a broader set of dense annotation benchmarks like Reefknot or previous related datasets enhance the robustness?

---

> ### Author Response · Authors · 2024-11-30
> **Response to Reviewer erAn**
>
> We sincerely thank Reviewer erAn for their thorough evaluation of our work and appreciate the opportunity to further clarify important aspects of our research. We apologize for the delay in our response and hope that our revisions address your concerns. Please feel free to reach out for further clarification or justification if needed.
>
> **W1, Q1, W3, Q3**. Need to evaluate two VISCON evaluation pipelines on more dense annotated datasets and more image data. Need to compare with more benchmarks.
>
> - Densely annotated datasets are less common, which presents a challenge for adding more such evaluations. To address this, we have utilized existing datasets such as VisualGenome and supplemented them with a new dataset created using oracle annotations from a 3D simulator. Therefore, we argue that we have evaluated two metrics on a decent broad range of densely annotated datasets. Moreover, our evaluation framework is built upon the mutual combination of dataset and metrics (evaluation pipelines).
>
> - Regarding the dataset size, our experiments have demonstrated that the current dataset is sufficient to effectively discriminate between different models (Appendix Figure 7). We observe that the evaluation metrics stabilize as the ratio of test data used increases, indicating that the dataset size is adequate for reliable and consistent model evaluation. Additionally, since our dataset is constructed without human annotators, scaling up is straightforward and cost-effective, but might be of diminishing returns as the dataset size increases.
>
> Quantitatively comparing our method to previous approaches is challenging due to the diversity of datasets and benchmarks, many of which are still in preprint form and not yet published (e.g., AMBER, PhD, FaithScore, Reefknot, CCEval, as reviewers indicate). According to the ICLR 2025 Reviewer Guide, it is not mandatory to discuss and compare with unpublished or very recent works. However, we can highlight key differences between our approach and several representative existing methods:
> - *FaithScore*: This method relies on visual entailment models, which are themselves are VLMs and prone to hallucinations, compromising their reliability. In contrast, our method uses dense human annotations to directly evaluate hallucinations, providing a more reliable and comprehensive assessment.
> - *CCEval*: This method extends reference object annotations using VisualGenome but still focuses primarily on objects and real-world photos like POPE, limiting its ability to capture the full range of visual concepts, such as attributes and relationships. Our method evaluates hallucinations across three types of visual concepts and multiple image domains, offering a more comprehensive evaluation of LVLM performance and investigation on the causes of hallucinations.
> - *AMBER, PhD and Reefknot*: These methods extend hallucination benchmarks to include attributes and relationships but face limitations. They either struggle with real-world scenarios involving multiple visual concepts due to their discriminative evaluation methods (only involving one concept per question) or lack dense annotations, leading to false negatives. Our method addresses these deficiencies by evaluating detailed caption responses, better discriminating LVLMs in complex tasks involving handling multiple concepts together, and enriching visual concept annotations with dense data from VisualGenome and oracle annotations from a 3D simulator, significantly reducing false negatives and enhancing reliability.
>
> Beyond the extensive visual concept types and annotation density that VISCON provides, our method uniquely explores the impact of image domains on LVLM hallucinations, a factor not addressed in previous research. Moreover, our experiments show that our method aligns more closely with human judgments, as demonstrated in the updated Table 5, highlighting its effectiveness and reliability in real-world scenarios.
>
> **W2, Q2**. Attributes might change after some stylizations. How VISCON handle these changes?
>
> We acknowledge that initially, our annotations did not fully account for changes in attributes after stylization. We have updated our annotations to address this oversight, manually removing color and material attributes from line painting and sketch-stylized images. After implementing these changes, we observed that the overall trend in model performance remained consistent, demonstrating the robustness of our evaluation framework.

---

### Official Review · Reviewer_QEkM · 2024-11-07

**Soundness:** 2
**Presentation:** 3
**Contribution:** 2
**Rating:** 5
**Confidence:** 4

**Summary:**

They introduce VISCON, a new benchmark framework aimed at evaluating vision hallucinations in Large Vision-Language Models (LVLMs) like GPT-4V and LLaVA. VISCON alleviates limitations of existing metrics by incorporating a diverse dataset with comprehensive visual concept annotations (objects, attributes, relationships) and includes two evaluation pipelines: an Earth Mover's Distance (EMD)-based approach for robust similarity assessment and an "Evaluate-By-Edit" pipeline for refining interpretability. Applying VISCON to six leading LVLMs, the study reveals insights into hallucination trends related to image domain shifts and response length, showing VISCON’s enhanced alignment with human judgments over existing methods.

**Strengths:**

1. VISCON includes a wide range of images across various styles and domains, enhancing the robustness and applicability of the evaluations.
2. Evaluating hallucination from multiple aspects of visual concepts—objects, attributes, and relationships—for a thorough assessment.
3. The paper introduces two metrics (EMD-based and “Evaluate-By-Edit” pipelines) for assessing hallucinations.

**Weaknesses:**

1. VISCON seems to directly use annotated scene graphs from VisualGenome and PROCTHOR. However, when the original images are stylized into other styles (e.g., line), the information may also change (e.g., the color of objects), which means the annotations should be adjusted accordingly. It should be clearly stated how the authors handled these changes or if they simply used the same annotations as in the original images.
2. In Table 2, there is too little differentiation in performance between models in terms of Earth Mover’s Distance (EMD). Some models (like GPT-4V) even outperform humans in the real-world original column of Table 2. However, GPT-4V still has many hallucination issues [1], which suggests this metric may not adequately capture the true performance gap between different models.
3. In lines 358-360 of the paper, the authors mention that there are some inconclusive queries in edit distances, though they believe these cases are rare and that edit distance still provides a reliable lower bound for estimating hallucination. This conclusion would benefit from further validation through qualitative or quantitative results to demonstrate the robustness of their metric.


[1] HALLUSIONBENCH: An Advanced Diagnostic Suite for Entangled Language Hallucination and Visual Illusion in Large Vision-Language Models. CVPR 2024

**Questions:**

Please kindly refer to weaknesses.

---

> ### Author Response · Authors · 2024-11-30
> **Response to Reviewer QEkM**
>
> We sincerely thank Reviewer QEkM for their thorough evaluation of our work and appreciate the opportunity to further clarify important aspects of our research. We apologize for the delay in our response and hope that our revisions address your concerns. Please feel free to reach out for further clarification or justification if needed.
>
> **W1**. Attributes might change after some stylizations. How VISCON handle these changes?
>
> We acknowledge that initially, our annotations did not fully account for changes in attributes after stylization. We have updated our annotations to address this oversight, manually removing color and material attributes from line painting and sketch-stylized images. After implementing these changes, we observed that the overall trend in model performance remained consistent, demonstrating the robustness of our evaluation framework.
>
> **W2**. EMD differs little. GPT-4V even surpass human on real-world photos.
>
> While the EMD differences might seem small on subsets of some image domains, they are quite significant when considering the entire dataset (Column 2 of Table 2). Moreover, these differences effectively distinguish between models and align well with human preferences (Table 5), underscoring their relevance and utility.
>
> Regarding GPT-4V's performance, it's possible that models tend to perform better on real-world images (those unaltered, natural photos). However, the real challenge—and where our research truly wants to reveal—is in evaluating performance on style-altered or abstract images. In these scenarios, models often struggle, especially on abstract images, similar to findings reported in HallusionBench. This finding also demonstrates the necessity of VISCON, which evaluates models on a diverse set of image domains like abstract and style-altered images.
>
> **W3**. Need to demonstrate the robustness of edit procedure.
>
> To address the need for demonstrating the robustness of the edit distance in our "Evaluate-By-Edit" pipeline, we have included qualitative results in Figure 9 of the appendix. This figure provides a detailed visualization of the edit process used to calculate the edit distance between the original model output and its revised version. In this visualization, we highlight text modifications with different colors to clearly show which parts of the text were removed, substituted, or inserted.
>
> As shown in Figure 9, the majority of the edits effectively correct vision hallucinations. These corrections specifically target references to entities that do not exist in the image, such as "a cat," "potted plant," and "a vase." Inconclusive contents from annotations, such as "some paintings appear to be smaller" is remained as-is.
> This visualization demonstrates how our method accurately identifies and corrects hallucinations, reinforcing the robustness of the edit distance as a measure in our evaluation pipeline.

---

### Official Review · Reviewer_8UyP · 2024-11-08

**Soundness:** 3
**Presentation:** 1
**Contribution:** 2
**Rating:** 5
**Confidence:** 4

**Summary:**

The paper addresses the issue of visual hallucinations in LVLMs by proposing a new evaluation dataset. This dataset spans various image styles and task types. Unlike traditional VQA formats that employ discriminative assessments, this work allows models to generate open-ended responses. To directly evaluate the quality of these generative outputs, the authors propose two new evaluation pipelines and metrics.

**Strengths:**

1. It explores the impact of different image styles on visual hallucinations.
2. The approach allows models to produce open-ended responses instead of simple yes/no answers. To enable evaluation within these generative outputs, two specialized pipelines and corresponding metrics have been designed.
3. The issue of visual hallucinations is a significant challenge for LVLMs, and research that reflects this problem from multiple perspectives and dimensions is valuable.

**Weaknesses:**

1. The motivation for creating a hallucination dataset needs clarification. This dataset does not specifically emphasize hallucination evaluation. While distinguishing between visual hallucinations and errors can be challenging, the paper should at least specify the targeted sources of hallucination rather than broadly adding data and task types. It’s important to clarify why the community needs this dataset.

2. A new hallucination dataset should provide novel insights. The lower performance on attributes and relationships compared to objects is expected and not unique to this dataset. More interesting conclusions, like the impact of domain shift and response length, are only briefly discussed. A deeper analysis of these factors is needed for a dataset paper.

3. It’s also worth noting that if the performance drop is simply attributed to insufficient training (L432-L434), it raises questions about the dataset’s significance. In fact, based on Table 2, the performance differences across image styles seem minimal, suggesting that adding relevant training data could easily address this issue.

**Questions:**

1. The link between image style and hallucinations needs further exploration. While a key contribution of this paper is testing hallucination across varied image styles, merely noting that style transformations (e.g., line drawings) cause performance drops is insufficient. Such drops are expected, given the limited training data for alternative styles, which seems more about training gaps than hallucinations. By contrast, POPE’s setup is more convincing, as it attributes hallucinations to textual context coexistence—a persistent issue regardless of training data expansion.

2. The paper lacks analysis on the sources of hallucinations. While POPE links hallucinations to high term co-occurrence and designs its pipeline around this, this paper's dataset mainly supplements existing data with new task types and styles. If the dataset only broadens existing types without addressing specific hallucination issues—especially if more training data could improve performance—its contribution may be limited.

3. The comparison with related work is insufficient. Some existing work, such as AMBER[1], PhD[2], FaithScore[3], has already expanded hallucination task types and evaluation formats, including attributes, relationships, and open-ended assessments. What is the unique contribution of your work compared to these existing studies?

4. The description of the two metrics designing and pipeline diagram is unclear. I recommend clarifying why both two metrics are needed and how they complement each other. Given the use of LLMs in the pipeline, why not directly use LLMs for end-to-end judgment?

The paper also lacks a clear presentation of the data format—images, prompts, and labels are hard to understand from text alone. Figure 1 devotes much space to visual hallucinations, which is widely understood, and may be unnecessary. Figures 2 and 3 are also hard to follow. For instance, in Figure 2(a), how is the sunflower image derived from transforming the motorcycle’s style? In L262, what is the meaning of the dissimilarity matrix? What is your specific computing method for d_{EDIT}?

Ref:
[1] An LLM-free Multi-dimensional Benchmark for MLLMs Hallucination Evaluation
[2] PhD: A Prompted Visual Hallucination Evaluation Dataset
[3] FaithScore: Fine-grained Evaluations of Hallucinations in Large Vision-Language Models

---

> ### Author Response · Authors · 2024-11-30
> **Response to Reviewer 8UyP**
>
> We sincerely thank Reviewer 8UyP for their thorough evaluation of our work and appreciate the opportunity to further clarify important aspects of our research. We apologize for the delay in our response and hope that our revisions address your concerns. Please feel free to reach out for further clarification or justification if needed.
>
> **W1**. Need to clarify the he motivation for creating the hallucination dataset, particularly regarding its specific focus on hallucination evaluation and the targeted sources of hallucination.
>
> The motivation for our dataset is driven by several key factors:
> - **Complexity of Visual Concepts**: Our dataset captures the complexity of visual concepts, particularly attributes and relations, where LVLMs are prone to hallucinate. Unlike previous methods such as POPE, AMBER, and CCEval, which often overlook these complex concepts, our dataset includes detailed annotations that require LVLMs to handle multiple components, such as object-subject relations and attribute-object pairs. This allows for a more accurate assessment of the true capabilities of different LVLMs.
> - **Dense Annotations with Minimal False Negatives**: We provide denser reference visual concepts by utilizing comprehensive annotations from VisualGenome and oracle annotations from a 3D simulator. Previous methods, like POPE and AMBER, often sample negative objects from non-annotated sets, leading to potential false negatives due to insufficient annotation density. As shown in the updated Table 1, our dataset achieves the highest density across all three types of visual concepts over previous annotation-based metrics, ensuring more reliable evaluations.
> - **Rigorous and Realistic Evaluation**: Our dataset offers a more rigorous and realistic evaluation of model performance by focusing on detailed captions that require LVLMs to handle multiple concepts simultaneously. This approach contrasts with simpler question-based benchmarks, providing a more comprehensive test of model capabilities in real-world scenarios.
>
> **W2, W3**. Need more discussions on domain shifts and length impact on hallucination. Small differences in EMD on different domain shifts.
>
> We have discussed the impact of domain shifts on hallucination in our paper in Section 4.2 of our paper. While the numerical range of EMD might appear relatively small, this does not diminish its ability to capture meaningful differences across domains, especially when comparing with human performances as shown in Figure 4(a). When comparing with human, a clear trend can be seen that LVLMs tends to generate more hallucinations as images deviate from real-world photos, particularly in more abstract domains like line paintings. This trend is consistent across models. Quantitatively, the EMD values increase as FID scores increase (Figure 4(b)), indicating that models struggle more with hallucinations as the visual domain becomes less conventional. This highlights the importance of evaluating models across diverse image types, which is a key contribution of our work.
>
> Regard on the impact of length, we found that the EMD values are consistent across different caption lengths (Figure 5(a)), indicating that the length of the caption does not significantly impact the hallucination levels. In Figure 5(b), we observe that edit distance linearly increase with caption length with a slope close to 1, indicating that the per-word hallucination rate level is not significantly impacted by the length of the caption, which is consistent with our observations on EMD.
>
> **W3, Q1, Q2**. The impact of image style to hallucination seems to be because of insufficient training. POPE attributes hallucination to textual coexistence in training corpus, which may not be solved by extending data. Adding new tasks and image styles in evaluations seems to be of limited contribution.
>
> It's important to note that while extending training data might reduce hallucinations over altered image styles, this does not eliminate the inherent deficiency in LVLMs. Unlike humans, who can well recognize visual concepts in artistic works despite seeing artistic works less frequently, LVLMs still struggle with response authenticity. Our findings highlight a gap in LVLMs that is not present in human cognition.
>
> Regard on the contribution of our dataset, we, and POPE both focused on uncommon data distributions, that leads to hallucinations for LVLMs. Ours focus on image domains (where some images may be less seen), and POPE on visual concepts' occurrences and co-occurrences (where some concepts or concept combinations are less seen). In principle, both of them could be solved by extending data. Moreover, our dataset adopt much denser annotations to minimize the presence of false negatives, which POPE tends to include more since it uses sampled negative visual concepts (in their work, object only) that are non-existent in annotations, which is much scarce and might sample objects exists in image, but un-annotated.

---

> > ### Author Response · Authors · 2024-11-30
> > **Response to Reviewer 8UyP (Continued)**
> >
> > **Q3**. Need discussion on the differences of previous work (e.g., AMBER, PhD, FaithScore, Reefknot).
> >
> > Quantitatively comparing our method to previous approaches is challenging due to the diversity of datasets and benchmarks, many of which are still in preprint form and not yet published (e.g., AMBER, PhD, FaithScore, Reefknot, CCEval, as reviewers indicate). According to the ICLR 2025 Reviewer Guide, it is not mandatory to discuss and compare with unpublished or very recent works. However, we can highlight key differences between our approach and several representative existing methods:
> > - *FaithScore*: This method relies on visual entailment models, which are themselves are VLMs and prone to hallucinations, compromising their reliability. In contrast, our method uses dense human annotations to directly evaluate hallucinations, providing a more reliable and comprehensive assessment.
> > - *CCEval*: This method extends reference object annotations using VisualGenome but still focuses primarily on objects and real-world photos like POPE, limiting its ability to capture the full range of visual concepts, such as attributes and relationships. Our method evaluates hallucinations across three types of visual concepts and multiple image domains, offering a more comprehensive evaluation of LVLM performance and investigation on the causes of hallucinations.
> > - *AMBER, PhD and Reefknot*: These methods extend hallucination benchmarks to include attributes and relationships but face limitations. They either struggle with real-world scenarios involving multiple visual concepts due to their discriminative evaluation methods (only involving one concept per question) or lack dense annotations, leading to false negatives. Our method addresses these deficiencies by evaluating detailed caption responses, better discriminating LVLMs in complex tasks involving handling multiple concepts together, and enriching visual concept annotations with dense data from VisualGenome and oracle annotations from a 3D simulator, significantly reducing false negatives and enhancing reliability.
> >
> > Beyond the extensive visual concept types and annotation density that VISCON provides, our method uniquely explores the impact of image domains on LVLM hallucinations, a factor not addressed in previous research. Moreover, our experiments show that our method aligns more closely with human judgments, as demonstrated in the updated Table 5, highlighting its effectiveness and reliability in real-world scenarios.

---

> > > ### Author Response · Authors · 2024-11-30
> > > **Response to Reviewer 8UyP (Continued)**
> > >
> > > **Q4**. Unclear motivations of two evaluation pipelines. Why not directly use LLMs?
> > >
> > > In previou methods, there are several straightforward method to evaluate hallucinations based on visual concept annotations.
> > > While using LLMs/VLMs to directly evaluate model response (e.g., FaithScore) against reference visual concepts might seem straightforward and simple for evaluating hallucinations, they have notable limitations. LLMs/VLMs themselves can introduce subjective errors in such complex task handling long caption and multiple visual concepts together, and often lack consistency across different samples, making it hard to distinguish between major and minor hallucinations (e.g., confusing "cat" with "dog" versus "computer monitor" with "laptop"). As a result, FaithScore only achieved a Pearson correlation of 0.48 while VISON achieved a much higher value of 0.95. Additionally, discriminative (i.e., QA-based) metrics (e.g., POPE, AMBER) that simply query LVLMs about the existence of specific visual concepts are too simplistic for real-world applications, where models need to handle multiple concepts simultaneously, such as in detailed captions or multi-turn questions.
> > > In contrast, despite more complex, our EMD-based approach offers several key advantages to address these deficiencies of current benchmarks: 1) we only abstract visual concepts with LLM, reducing ambiguity and simplifying the task for LLMs, which only require LLMs to repeat specific words from the input. 2) we use specialized sentence embedding models that are better suited for discriminating text pairs, providing nuanced, soft similarity scores rather than binary 0/1 values. 3) we evaluate LVLM's hallucinations through their long response when required to describe image in details, better discriminating different models with this more complex task.
> > > By delegating the evaluation to two simpler tasks and specialized models—concept extraction and similarity discrimination—we enhance reliability and ensure intra-sample consistency, which end-to-end LLM-based scores often lack. By evaluating through detailed captions, our metric better discriminate different LVLMs.
> > > Our benchmark's effectiveness is further validated by human preference alignment experiments, as shown in Table 5, demonstrating its superiority over previous methods and simpler QA-based evaluations.
> > > On the other hand, our edit distance based pipeline aims to provide interpretability in hallucination evaluations by highlighting the specific changes needed to align model outputs with a refined version.
> > >
> > >
> > > **Q5**. Figures needs improvement. Tech details are missing.
> > >
> > > We acknowledge that Figure 2(a) currently does not display the corresponding image stylizations. The sunflower image shown is derived from a real-world photo of a sunflower. We have provided a more clear comparison in Figure 10 in appendix, and we will address this issue in main paper and ensure that the final version includes the appropriate image stylizations for clarity.
> > >
> > > Regarding the technical details: In line 262, the dissimilarity matrix is calculated as (1 - similarity matrix). This is because it represents the embedding distance between two sets of visual concepts. To compute $d_{E D I T}$, the edit distance, we first split the two sentences into word sequences. We then calculate the edit distance between these sequences using the NLTK package's tokenization and edit distance toolkit.
> > >
> > > We appreciate the feedback and will make these improvements in the final version to enhance clarity and accuracy.

---

### Author Response · Authors · 2024-11-28
**General Response to All Reviewers**

We thank all reviewers for their thorough evaluation of our work and recognizing the comprehensive nature of our hallucination benchmark (Reviewers 8UyP, QEkM, erAn, 58J2) and broad coverage of visual concepts including objects, attributes, and relationships (Reviewers QEkM, erAn). Meanwhile, the reviewers acknowledged our methodological contributions, specifically on our novel evaluation metrics and pipelines (Reviewers QEkM, erAn, 58J2) and the shift toward open-ended responses rather than simple yes/no answers (Reviewer 8UyP). Reviewers also appreciated the thorough experimental validation (Reviewers 58J2, kDMt), clear presentation and motivations (Reviewer kDMt), and strong alignment with human evaluation (Reviewer erAn).

We appreciate the reviewers' suggestions and have carefully revised our paper accordingly. Our major revisions include:
1. More discussions of previous hallucination benchmarks in the related work section (L157-158), with updated comparisons on annotation density (Table 1) and human preference alignment experiments (Table 5)
2. Removal of colors and material annotations in sketch and line painting stylized images, with corresponding updates to evaluation results (Table 2 and 5)

All revisions are marked in purple (figure and table modifications are marked by colorizing captions) in the updated PDF for easy reference.

---

### Meta-Review · Area_Chair_4R6g · 2024-12-20

**Metareview:**

The paper introduces a new benchmark framework, VISCON, and tries to evaluate the vision hallucination issue in large vision-language models (LVLMs), such as GPT-4V and LLaVA. It consists of two pipelines: an Earth Mover's Distance (EMD)-based method for distributional similarity and an Evaluate-By-Edit approach to gauge response alignment with annotated visual concepts.

Strengths:
+ The method for data collection and evaluation is intuitive. The benchmark is curated comprehensively by considering various image domains, styles, and dense annotations.
+ VISCON aligns more closely with human preferences than other metrics.

Weaknesses:
+ The motivation needs to be further clarified. And the comparisons with related hallucination work are limited.
+ The comparisons with dense annotated benchmarks are missing.

**Additional Comments On Reviewer Discussion:**

During the rebuttal, some concerns are been addressed, such as: the motivation for proposing new metrics for detecting hallucinations.

After the rebuttal, the submission received **all negative ratings**. The main concerns are: 1) The training and evaluation procedures are not convincing  (such as too complex procedures). 2) More comprehensive experimental results are expected.

---

### Decision · Program_Chairs · 2025-01-22

Reject